# Highly reversible transition metal migration in superstructure-free Li-rich oxide boosting voltage stability and redox symmetry

Tianwei Cui[1,6], Jialiang Xu[2,6], Xin Wang[1], Longxiang Liu[3], Yuxuan Xiang[4,5], Hong Zhu[2], Xiang Li[1] ✉ & Yongzhu Fu[1] ✉

The further practical applications of Li-rich layered oxides are impeded by voltage decay and redox asymmetry, which are closely related to the structural degradation involving irreversible transition metal migration. It has been demonstrated that the superstructure ordering in O2-type materials can effectively suppress voltage decay and redox asymmetry. Herein, we elucidate that the absence of this superstructure ordering arrangement in a Ru-based O2-type oxide can still facilitate the highly reversible transition metal migration. We certify that Ru in superstructure-free O2-type structure can unlock a quite different migration path from Mn in mostly studied cases. The highly reversible migration of Ru helps the cathode maintain the structural robustness, thus realizing terrific capacity retention with neglectable voltage decay and inhibited oxygen redox asymmetry. We untie the knot that the absence of superstructure ordering fails to enable a high-performance Li-rich layered oxide cathode material with suppressed voltage decay and redox asymmetry.

In the last decade, Li-rich layered oxide cathode materials attracted tremendous attention due to the elevated specific capacity of more than 250 mAh g$^{-1}$ combined with the redox reactions of cations and anions, which favors the development of high-energy-density batteries[1–7]. For Li-rich layered oxides, the formation of Li-O-Li configuration would create a new oxygen band with increased energy overlapped with the transition-metal (TM) d band, which makes the oxygen redox reaction easy to occur for charge compensation[8–10]. However, oxygen redox is a double-edged sword. In addition to the sluggish kinetics, the asymmetric oxygen redox also results in a low initial coulombic efficiency and rapid drop of voltage. Besides, the lattice oxygen release is always accompanied by the formation of the in-plane rearrangement and even out-of-plane migration of TM ions[11,12]. It is believed that the voltage decay is primarily coupled with irreversible out-of-plane TM migration[13,14], which makes the migrated TM ions partially occupy the sites of Li, resulting in the gradual formation of spinel-like phase with low output voltage, especially for O3-type

oxides[15,16]. Inspiringly, the recent reports claimed that the ribbon-type superstructure would inhibit the structural disordering resulting from the TM migration, which significantly enhances the reversibility of oxygen redox reaction and suppresses voltage loss[14,17,18]. These studies strongly unraveled the close relationship between superstructure ordering and structural stability.

Attractively, the previous reports have demonstrated that the conversion from layered to spinel would be restricted in O2-type oxides, providing a guideline to reduce voltage decay[13,19–23]. For example, Xia's group reported a new O2-type Mn-based oxide Li$_{1.25}$Co$_{0.25}$Mn$_{0.50}$O$_2$ with partial superstructure ordering, which delivers a reversible capacity of ~400 mAh g$^{-1}$ with suppressed voltage fade[20]. Innovatively, the mechanism of inhibiting voltage decay in the O2-type oxide is uncovered unequivocally by previous work[21]. The authors demonstrated that the further in-plane TM migration in the alkali metal (AM) layer would be effectively restricted in an O2-type structure with superstructure ordering, which conducively

[1]College of Chemistry, Zhengzhou University, Zhengzhou 450001, China. [2]University of Michigan-Shanghai Jiao Tong University Joint Institute, Shanghai Jiao Tong University, Shanghai 200240, China. [3]Department of Materials, University of Oxford, Parks Road, Oxford OX1 3PH, UK. [4]Research Center for Industries of the Future, Westlake University, Hangzhou, Zhejiang 310030, China. [5]School of Engineering, Westlake University, Hangzhou, Zhejiang 310030, China. [6]These authors contributed equally: Tianwei Cui, Jialiang Xu. ✉e-mail: xli@zzu.edu.cn; yfu@zzu.edu.cn

streamlines the returning path and increases the reversibility of migrated TM. Nonetheless, even though voltage decay has been greatly inhibited in O2-type oxides, there are still some other factors closely associated with the irreversible TM migration[24-26]. More recently, the study revealed the importance of cations superstructure ordering on voltage decay and oxygen redox stability[24]. They showed that the absence of superstructure ordering would make an irreversible TM migration occur easily, thereby leading to gradual voltage decay and oxygen redox asymmetry.

In this study, we focus on a superstructure-free Li-rich Ru-based system to disclose if the superstructure ordering is necessary to achieve highly suppressed voltage decay involving reversible TM migration. Two premises are considered first, the one is that Ru-based systems play a key role in elucidating oxygen redox and voltage decay issues, and the strong covalence of Ru-O enables the oxygen redox more reversible[27-30]. The other is that O2-type configuration is structural robustness inherently[13,21,23], which can act as a suitable platform to help comprehend the voltage decay in other oxide systems containing oxygen redox. From this respect, we originally designed a superstructure-free O2-type Li-rich Ru-based layered oxide $Li_{0.6}Li_{0.2}Ru_{0.8}O_2$ (denoted as O2-LLRO hereafter). Surprisingly, a very different TM migration path with high reversibility is observed, enabling the cathode a high voltage stability and redox symmetry. The mechanism is carefully investigated by various advanced characterizations. Profited from high structural stability originating from the reversible TM migration, the cathode delivers stable capacity retention (90.2% after 100 cycles) and neglectable voltage decay (97.7% voltage retention after 100 cycles). Our findings confirm that the reversible TM migration can still be achieved in the superstructure-free layered oxides, which vastly enhances the whole structural robustness and oxygen redox symmetry.

## Results

### Structure of O2-LLRO

The superstructure-free O2-LLRO was obtained by an ion exchange process from $P2-Na_{0.6}Li_{0.2}Ru_{0.8}O_2$. The precursor $P2-Na_{0.6}Li_{0.2}Ru_{0.8}O_2$ was synthesized by the traditional solid-state method, which can be well indexed as P2-type with a space group of $P6_3/mmc$ from the XRD pattern (Supplementary Fig. 1), with corresponding refinement results in Supplementary Table 1. The as-prepared LLRO was characterized by XRD tests and refined by GSAS-II (Fig. 1a)[31]. Note that the stacking faults and defects resulting from the ion exchange process weaken the intensity of diffraction peaks and give rise to a weak peak at around 21° [32]. The corresponding refinement results indicate the formation of a layered structure with a space group of $P6_3mc$, indexed as the O2-type structure with "ABAC" oxygen stacking sequence in one unit, in which $Li^+/Ru^{4+}$ ions occupy the octahedral sites in TM layer and the other $Li^+$ ions remain in AM layer (Fig. 1b)[33]. Therefore, the crucial Li-O-Li configuration is formed, which has been proven a trigger to induce the oxygen redox (Fig. 1a inset)[1,34]. The refined lattice parameters of LLRO are $a = b = 2.8375$ Å, $c = 9.6564$ Å, $V = 69.33$ Å$^3$, $\alpha = \beta = 90°$ and $\gamma = 120°$ (Supplementary Table 2).

High-angle annular dark-field scanning transmission electron microscopy (HAADF-STEM) has been widely used to observe the local atom structure[14,35]. The images show that LLRO presents a superstructure-free disordered structure (Fig. 1c). The O2-type layered structure of LLRO was also evidenced by HAADF-STEM, marked by the yellow area. The bright dots represent the heavy metal Ru in the TM layer, and Li cannot be detected in this model. What's more, the distance of two adjacent TM layers is measured as 0.517 nm, which is consistent with the distance obtained from (002) plane in Fig. 1a. The distance of two adjacent Ru atoms is measured as 0.260 nm. Integrated differential phase contrast STEM (IDPC-STEM) and annular bright field STEM (ABF-STEM) were further adopted to verify the atoms arrangement of LLRO directly (Fig. 1d, e), by which the light atoms such as Li and O can be observed[36]. In Fig. 1d, the green, red, and yellow circles represent Li, O, and Ru atoms, respectively. It is clearly seen that Li atoms are located in the Li layer, and Li/Ru atoms are located in the TM layer with the superstructure-free arrangement, bridged by O atoms, corresponding to a typical layered structure. ABF-STEM image (Fig. 1e) further shows the TM layer and AM layer, represented by the yellow and green arrows, respectively. What's more, the selected area electron diffraction (SAED) pattern shows the presence of (002), (004), (006), (102), and (103) planes along the [1$\bar{1}$0] zone axis, indicating the diffraction patterns of the O2-type structure (Fig. 1f). The stacking faults are also visible in the SAED pattern[32]. Besides, SEM image of LLRO powder displays the typical layered structure and uniform particle morphology with an average diameter

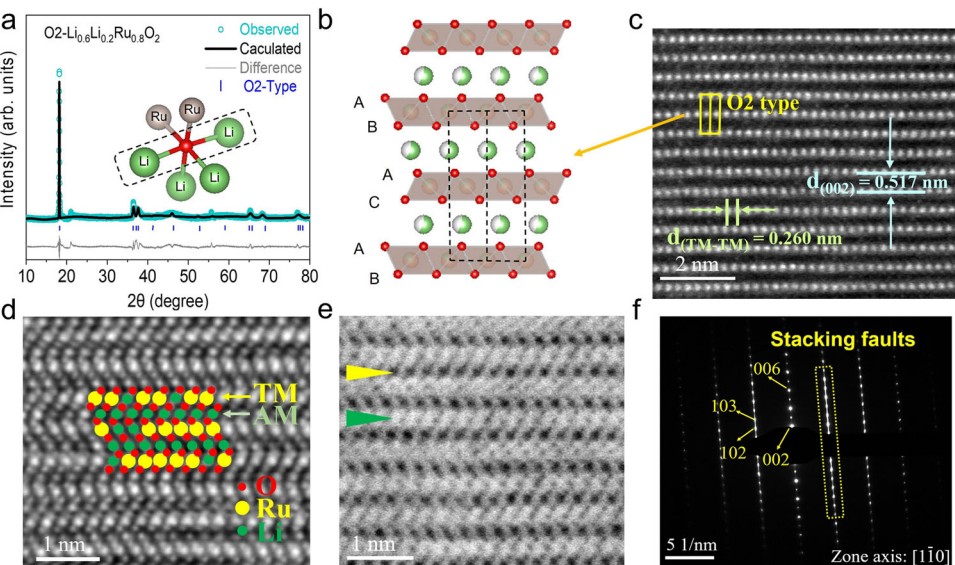

**Fig. 1 | Structural characteriazations of LLRO. a** XRD pattern of LLRO with Rietveld refinement. (inset: Li-O-Li configuration). **b** Schematic O2-type crystal structure. **c** HAADF-STEM image, **d** IDPC-STEM image, and **e** ABF-STEM image of LLRO along the [1$\bar{1}$0] zone axis. **f** SAED patterns of LLRO along the [1$\bar{1}$0] zone axis. Source data are provided as a Source Data file.

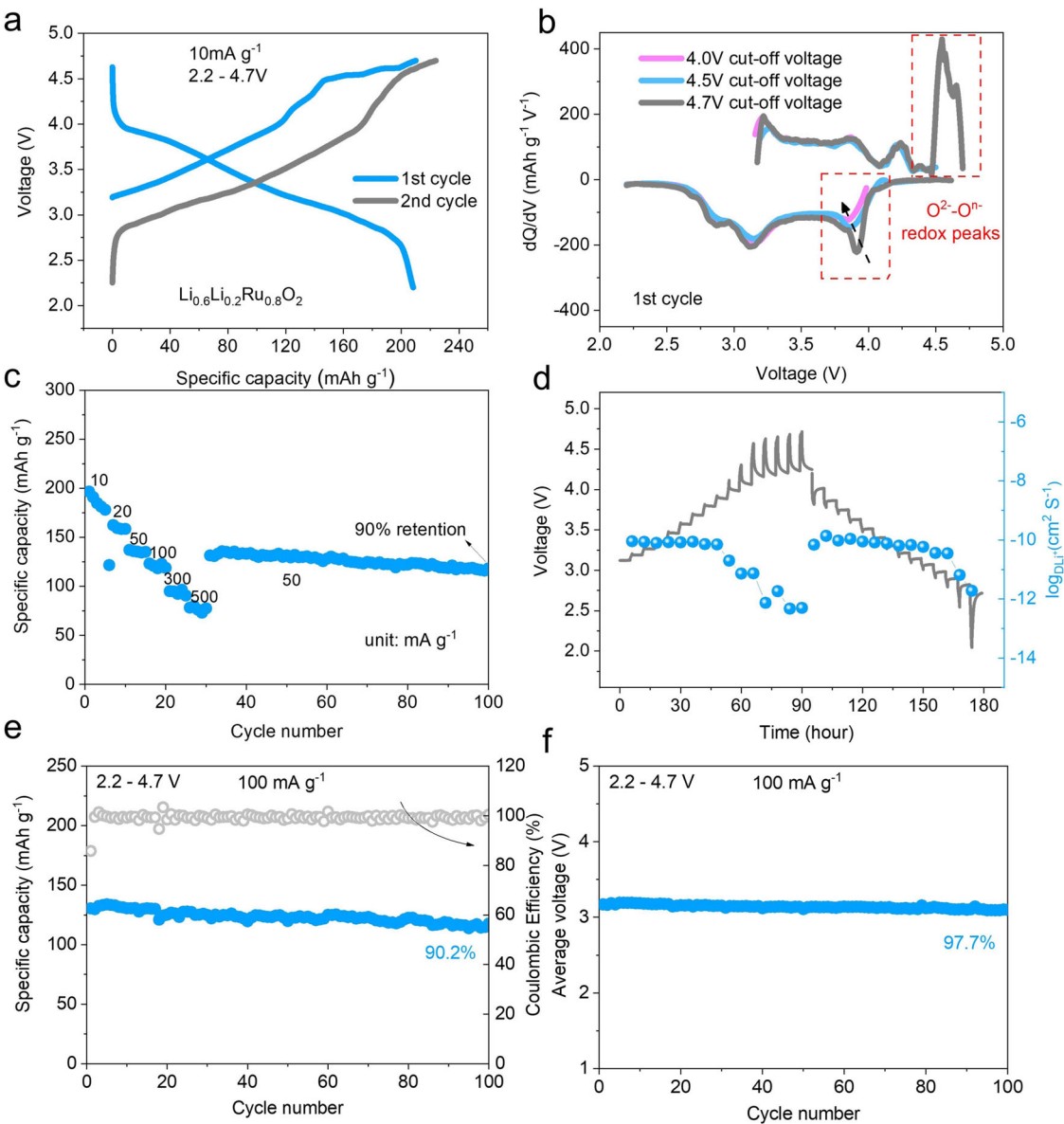

**Fig. 2 | Electrochemical performance of LLRO. a** The initial two charge/discharge curves at 10 mA g⁻¹ between 2.2–4.7 V. **b** The dQ/dV curves at different cut-off voltage windows. **c** Rate performance. **d** GITT curves for the first cycle with calculated Li⁺ diffusion coefficient. **e** Long-term cycling performance at 100 mA g⁻¹. **f** Average potential during cycling at 100 mA g⁻¹. Source data are provided as a Source Data file.

of ~2-10 μm (Supplementary Fig. 2). The above results unambiguously confirm the acquisition of the O2-type layered material LLRO with disordered Li/Ru arrangement.

**Electrochemical performance**

Figure 2a displays the initial charge-discharge curves of LLRO as a cathode in the voltage of 2.2–4.7 V at 10 mA g⁻¹, showing the initial discharge capacity of 208 mAh g⁻¹ with a typical S-shape trend corresponding to around 0.9 mol Li⁺ reinsertion. The ABF-STEM images of the de-lithiated state and re-lithiated state are shown in Supplementary Fig. 3, revealing the removal and insertion of Li⁺ during cycling. Similar to many other Li-rich materials such as $Li_{1.2}Ni_{0.13}Co_{0.13}Mn_{0.54}O_2$, $Li_{1.2}Ni_{0.2}Mn_{0.6}O_2$, and $Li_2RuO_3$[9,37,38], the first charge curve can be divided into two parts represented by a slope below 4.5 V and then a plateau above 4.5 V, which is more visualized in the dQ/dV analysis (Fig. 2b). The capacity delivered within the slope region might be contributed by the partial oxidation of $Ru^{4+}$ while the high voltage plateau might be associated with oxidation of oxygen. During the

discharge, there are two obvious peaks in the dQ/dV curves. The peak at the high voltage region belongs to the reduction of oxygen species, while another roughly corresponds to the reduction of the cation. More importantly, when the cut-off voltage decreases to 4.5 V and 4.0 V (Supplementary Fig. 4), there is an obvious drop in the peak corresponding to the reduction of oxygen species. The peak assigned to the cationic reduction shows very mild changes (Fig. 2b), proving the high symmetry and reversibility of oxygen redox. The rate performance of LLRO from 10 mA g⁻¹ to 500 mA g⁻¹ is shown in Fig. 2c. When the rate recovers to 50 mA g⁻¹, a reversible capacity of more than 130 mAh g⁻¹ can still be maintained, with excellent retention of 90% after the next 70 cycles. Galvanostatic intermittent titration technique (GITT) was used to verify the diffusion kinetics of Li⁺ (Fig. 2d)[39]. The detailed parameters are marked in Supplementary Fig. 5. The calculated values of $D_{Li+}$ are displayed as blue dots, which range from $10^{-10}$ to $10^{-13}$ cm² s⁻¹, showing the decent kinetics of LLRO. Nonetheless, the diffusion coefficient presents a sharp fall at the high voltage regions, correlated with the oxidation of oxygen, conforming to the poor

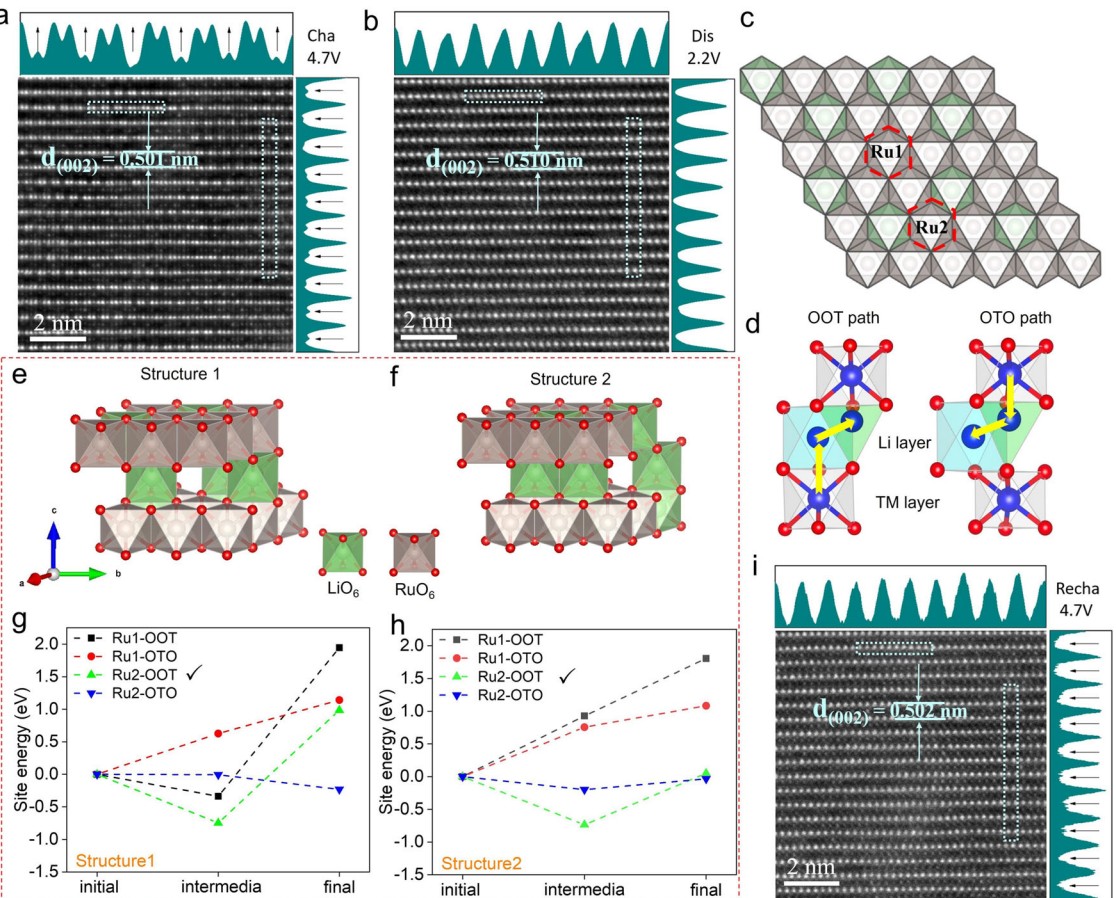

**Fig. 3 | TM migration phenomenon in LLRO. a** HADDF-STEM image of the superstructure-free LLRO at the 4.7 V charged state and **b** 2.2 V discharged state along the [1$\bar{1}$0] zone axis. The graphs on the right and top are the HAADF signal profiles of LLRO. The arrows in the signal profiles indicate the variation of TM and vacancy. **c** Structure diagram of TM layer for O2-LLRO. **d** Two migration paths of Ru in O2-LLRO (O: octahedron site, T: tetrahedron site). **e, f** Two different structures at the half-charged state. The calculated relative site energy along the migration path for Ru1 and Ru2 in **g** structure 1 and **h** structure 2. **i** HADDF-STEM image of LLRO at the 4.7 V recharged state. Source data are provided as a Source Data file.

kinetics of anionic oxidation, which will be discussed in the later part. As well, benefited from the reversible oxygen redox and structure evolution, the LLRO cathode can exhibit a superior capacity retention of 90.2% after 100 cycles at 100 mA g$^{-1}$ between the voltage window of 2.2–4.7 V (Fig. 2e). More importantly, the average discharge potential retains up to 97.7% after 100 cycles (0.73 mV per cycle) (Fig. 2f). In addition, the normalized discharge curves of O2-LLRO show neglect-able voltage drop which is obvious in O3-Li$_2$RuO$_3$ upon cycling (Supplementary Fig. 6). Considering that voltage drop is mainly associated with the irreversible TM migration, it can be predicted that O2-LLRO may retain a structural evolution involving reversible TM migration during cycling. The good capacity and voltage retention motivate us to uncover the corresponding cation migration and structure evolution of LLRO in the later part.

**Cation migration**

To confirm the local structural evolution of LLRO, HAADF-STEM images accompanied by Fourier transform were collected along the [1$\bar{1}$0] zone axis at the different states. Figure 3a shows slight Ru migration from the octahedral site of the TM layer (O$_{TM}$) to the octahedral site of the AM layer (O$_{AM}$) during the first de-lithiation process, together with the decrease of the width of two adjacent TM layers (0.501 nm) due to a large amount of the Li$^+$ removal. Especially, it should be highly aware that the dumbbell-like honeycomb ordering is formed at the fully-charged state (Fig. 3a), which is quite different from the pristine disordered structure (Fig. 1c–e). The variation must be related to the out-

plane displacement of Li in the TM layer (Li$_{TM}$) and the out-plane migration of Ru. At the high voltage regions, Li$_{TM}$ begins to migrate toward the AM layer once Li in the AM layer (Li$_{AM}$) has been depleted, together with out-plane migration of Ru leaving some vacancies in the TM layer. Upon the re-lithiation process, the HAADF-STEM image of O2-LLRO expresses a reversal in comparison with that of de-lithiation process, represented by the rearrangement of the cations situated in the TM layer. Distinctly, nearly all of the migrated cations move back accompanied by the slight expansion of the AM layer (0.510 nm), and there are no longer bright dots in the AM layer (Fig. 3b). Meanwhile, the re-insertion of migrated Ru cations fills the vacancies emerging at the fully-charged state, indicating the perfect reversibility. The supposed Li displacement will be confirmed by $^7$Li solid-state NMR in the later section.

First-principles calculations were conducted to verify the Ru migration phenomenon firstly, with the optimized structure shown in Supplementary Fig. 7. Note that there are two kinds of Ru atoms in O2-LLRO, where Ru1 is surrounded by TM ions while Ru2 is adjacent to Li and TM ions (Fig. 3c). Figure 3d shows the schematic diagram of the migration paths of Ru in O2-LLRO, in which there are two migration paths for Ru ions profited from the face-shared arrangement between RuO$_6$ in O$_{TM}$ and LiO$_6$ in O$_{AM}$. For the O2-type structure, Ru ions could migrate from O$_{TM}$ sites to the nearest O$_{AM}$ sites first and then continue to the tetrahedral Li site (T$_{AM}$) during the charging process (denoted as OOT path hereafter). Meanwhile, Ru ions could also migrate through the O$_{TM}$-T$_{AM}$-O$_{AM}$ path (denoted as OTO path hereafter).

We comparatively calculated the site energies of the intermediate and final sites considering all the possible migration cases of Ru.

It is still worth noting that there are two structures at the half-charged state, whose schematic illustrations are listed in Fig. 3e, f. For structure 1, Fig. 3g shows the migration to the intermedia site for Ru2 along the OOT path is thermodynamically feasible in the four considered cases, of which the site energy is −0.75 eV (value details in Supplementary Table 3). The final site energy is 0.99 eV, indicating that Ru2 prefers to migrate from $O_{TM}$ to $O_{AM}$ during the initial charge process, but the further in-plane migration to $T_{AM}$ will be significantly inhibited. For structure 2, Ru1 is still prone to be immovable in the de-lithiated state because of the higher intermedia site energies (0.93 eV for OOT path and 0.76 eV for OTO path), as shown in Fig. 3h and Supplementary Table 4. Nonetheless, the lowest intermedia site energy manifests that the out-plane migration from $O_{TM}$ to $O_{AM}$ of Ru2 is likely to occur, but the further in-plane migration to $T_{AM}$ site requires a thermodynamic penalty of approximately 0.79 eV.

To sum up, the above theoretical calculation results demonstrate that Ru2 in O2-LLRO tends to migrate from $O_{TM}$ to $O_{AM}$ along the OO path (not OTO path) and stays at the $O_{AM}$ site finally. The migrated TM ions can readily return to the pristine sites upon the re-lithiation process, in line with the observed HAADF-STEM image (Fig. 3b). Nevertheless, for the typical O3 materials, such as O3-$Li_2RuO_3$, Ru ions would only migrate through the OTO path during the charge process due to the edge-shared arrangement between $RuO_6$ in $O_{TM}$ and $LiO_6$ in $O_{AM}$, and stay at $O_{AM}$ sites finally[27,30,40]. The fussy OTO migration and slab gliding of O3 to O1 would inevitably leave the partial migrated Ru trapped in the AM layer during the discharge process, which is unfavorable for the Li+ (de)insertion, leading to an inferior capacity retention compared with that of O2 ones after 100 cycles (Supplementary Fig. 8). When recharging to 4.7 V, the bright dots reappear in the AM layer with the minor shrink, demonstrating the presence of reversible Ru migration (Fig. 3i). Importantly, the honeycomb superstructure ordering does not appear at the fully recharged state, which will be further discussed later. To further verify the reversibility of TM migration, the HAADF-STEM of O2-LLRO after 10 cycles was also performed. As shown in Supplementary Fig. 9, there are almost no TM ions left in the Li layer, showing a highly reversible TM migration of O2-LLRO, which provides a solid structural foundation for splendid capacity and voltage retention.

$^7Li$ solid-state NMR was performed to monitor the local environment evolution of Li (Fig. 4a and Supplementary Fig. 10), with schematic illustrations shown in Fig. 4b to better comprehend the evolution. At the pristine state, there are two kinds of Li located at 53 ppm and 32 ppm with a ratio of 1:3, corresponding to $Li_{TM}$ and $Li_{AM}$, respectively. When charging to 4.5 V, nearly all $Li_{AM}$ (~0.6 mol) has been deintercalated, and $Li_{TM}$ remains immobile in the TM layer. Further lifting voltage to 4.7 V makes most of $Li_{TM}$ hop into the AM layer first and then escape from the lattice to contribute to capacity, leaving a very small amount of Li still staying in the TM layer and AM layer. Upon the re-lithiation, 0.9 mol Li (according to the discharged specific capacity in Fig. 2a) reinserts into the lattice and only occupies the AM layer because the AM layer is enough to accommodate no more than 1 mol Li, and there is no longer $Li_{TM}$ according to the NMR results.

Here, we make a brief summary of the migration characteristic of Ru in O2-LLRO. For the pristine state, the structure is disorder (Fig. 1a), which is similar to the recently reported O2-type superstructure-free $Li_x(Li_{0.25}Co_{0.25}Mn_{0.5})O_2$ material (denoted as O2-LLCM hereafter)[24]. Theoretical calculation results indicate that some TM ions of O2-LLCM could migrate through the OTO path and occupy $O_{AM}$ sites finally. The multistep return path makes the partially migrated TM ions still occupy the AM layer according to the HAADF-STEM image after an electrochemical cycle[24]. The irreversible TM migration inevitably results in the gradual voltage fade and capacity decrease. Differently, Ru ions in O2-LLRO are more likely to migrate through the OO path and

stay at intermedia $O_{AM}$ sites according to the theoretical calculation results (Fig. 3g, h), the single-step return path makes Ru migration more reversible (Fig. 3b). We propose that the path discrepancy may be associated with the size mismatch. There is a higher thermodynamic barrier for Ru ions with a larger atom radius to pass through the narrow intermedia tetrahedra. As a result, the bigger octahedron is an alternative. While the intermedia tetrahedra is enough for the smaller Mn ions to pass through.

On the other hand, the formation of honeycomb ordering at the fully charged state (Fig. 3a) must have a close relationship with the out-plane migration of $Li_{TM}$ and Ru. However, the honeycomb ordering does not reappear at the fully recharged state even though the out-plane migration of Ru still occurs at that state (Fig. 3i). According to the $^7Li$ solid-state NMR results, at the outset of the second cycle (discharged-2.2 V state of the first cycle), there are no longer Li ions in the TM layer (Fig. 4a). Therefore, we propose that the absence of $Li_{TM}$ should be responsible for the failure to emergence of honeycomb ordering structure in the second charge state, which is worth studying in the future reports. Although the structure of O2-LLRO during the initial charge process evolves into the superstructure partially, which may be significantly associated with the out-of-plane migration of $Li_{TM}$ ions. It should be emphasized that the superstructure does not exist in the important pristine state, the initial discharged state, and thereafter the extended cycles. For these reasons, O2-LLRO is regarded as a superstructure-free material.

## Structural stability

Moreover, in situ XRD experiment was conducted to obtain the phase evolution of LLRO (Supplementary Fig. 11). As shown in Fig. 4c, during the initial de-lithiation process, the (002) peak gradually shifts towards the higher angle without obvious phase variation, meaning the minish of c-lattice parameter, which is analogous to other O2-type materials[22,24]. The minish may stem from the decrease of ion radius caused by the oxidation of $Ru^{4+}$ or/and lattice $O^{2-}$. Next, the (002) and (004) peaks move back to the lower angle close to the pristine state upon the initial discharge process (Supplementary Fig. 11), indicating that a similar amount of Li re-inserts into the AM layer. Upon the recharging process, the evolution trend is similar to that of the initial de-lithiation process, revealing a reversible process. The reversible structural evolution is accessible to excellent long-term performance. In addition, other than the mild shifts, no extra phase changes appear, revealing the great reservation of the O2-type structure during the cycling process. Ex situ XRD patterns of O2-LLRO exhibit proximate phase evolution with that of the in situ experiments (Supplementary Fig. 12). In contrast, O3-$Li_2RuO_3$ is subjected to the sequential three-phase changes of O3-O1'-O1 during the initial charge process (Fig. 4d and Supplementary Fig. 13), similar to the previous reports[41,42]. Predictably, long-range phase changes inevitably worsen cell performance[43,44]. Importantly, even after 100 cycles, the O2-type layered structure is still preserved, demonstrating that the LLRO cathode displays robust structural stability (Fig. 4e). As a sharp contrast, O3-$Li_2RuO_3$ suffers from severe structure distortion after 100 cycles (Fig. 4f). The XRD pattern shows not only the broader and lower intensity of (003) peak associated with the decrease of grain size and crystalline, but also with several peaks (dis)appearing. The severe structure collapse may result from the successive irreversible TM migration[27,30,45].

Raman analysis was performed to investigate the change of bonding character of LLRO, which has been proven to be effective in detecting the local bonding evolution from layered to spinel structure[46]. The Raman spectra of the pristine and cycled samples of LLRO are displayed in Fig. 4g. For the pristine, there exist two peaks at 609 $cm^{-1}$ and 493 $cm^{-1}$, which are described as $A_{1g}$ with symmetrical stretching and $E_g$ with symmetrical deformation of TM-O bond, respectively[46]. Obviously, all peaks are preserved even after 50 cycles.

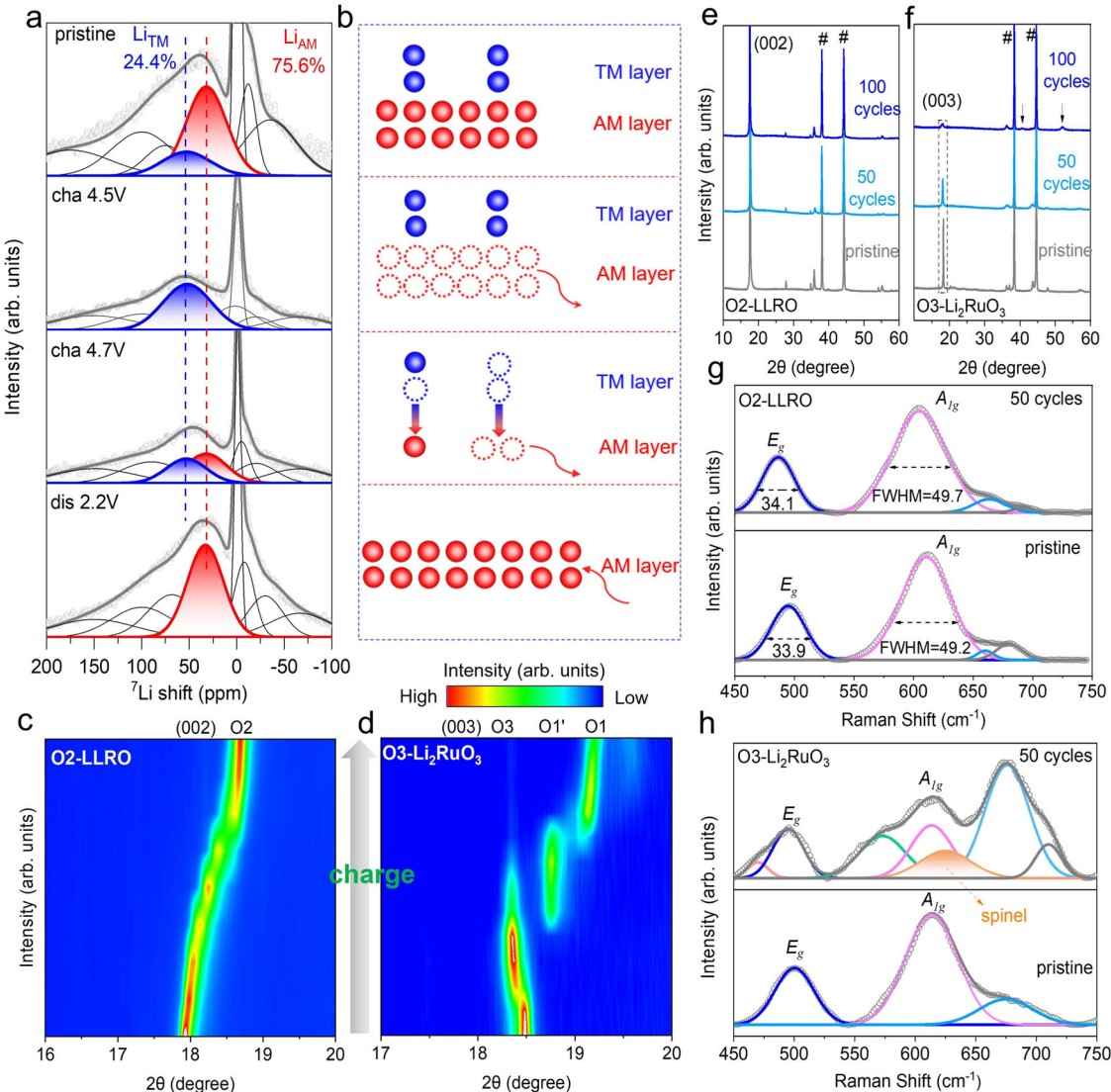

**Fig. 4 | Structural evolution of LLRO and Li₂RuO₃. a** ⁷Li solid-state NMR spectra (cha: charge; dis: discharge) and **b** corresponding schematic diagram of the superstructure-free LLRO at the different states. Contour maps of in situ XRD for **c** (002) peak of O2-LLRO and **d** (003) peak of O3-Li₂RuO₃ at the initial charge process. Ex situ XRD patterns of **e** O2-LLRO and **f** O3-Li₂RuO₃ after cycles. Ex situ Raman spectra of **g** O2-LLRO and **h** O3-Li₂RuO₃ at pristine and after 50 cycles. The orange arrow represents the spinel phase. Source data are provided as a Source Data file.

In detail, the positions and full width at half maximum of the cycled sample show negligible variations compared with that of the pristine, indicating the good stability of the local bonding character. However, O3-Li₂RuO₃ shows considerable structural variation after 50 cycles (Fig. 4h), agreeing well with the results of ex situ XRD (Fig. 4f). More importantly, there emerges an extra peak at 625 cm⁻¹ relevant to the formation of the spinel-domain in O3-Li₂RuO₃[47], which clearly contrasts with the absence of that in O2-LLRO, reflecting that the transformation from layered to spinel phase is well restricted in O2-LLRO, enabling the superior voltage retention (Supplementary Fig. 8).

### Charge compensation mechanism

To further investigate the cationic and anionic redox behaviors of the O2-LLRO cathode, Ru K-edge X-ray absorption near edge structure and O K-edge soft X-ray absorption spectroscopy (sXAS) were collected. Figure 5a shows that the Ru absorption energy shifts towards higher energy upon the initial charge process, suggesting the oxidation of Ru during the Li⁺ extraction. The rising energy moves back to the original position after the subsequent discharge, indicating the reduction of oxidized Ru. Extended X-ray absorption fine structure (EXAFS) spectra

and wavelet-transformed (WT) EXAFS spectra were also employed to present the variations of the local coordination environment (Fig. 5b, c). There are two major signals in the Fourier-transformed spectra, one centered at ~1.5 Å belonging to the Ru-O bond and the other located at more than 2.5 Å representing the Ru-Ru coordination. Note that the positions of pristine Ru-O and Ru-Ru bond coincide with that of RuO₂ standard sample, attesting the valence state of Ru in pristine LLRO is +4. The intensity of Ru-O in the first shell reduces obviously after the first charge, which might be associated with an increased disorder of Ru-O coordination and/or the formation of oxygen vacancies generated from the Li⁺ removal. Besides, the movement of Ru-O distance towards the lower distance provides solid evidence of the oxidation of Ru[48]. More importantly, the peak of Ru-O reverts to its pristine states for both the distance and intensity after the initial cycle, proving the high reversibility of the local chemical environment in LLRO. In addition, the Ru-Ru peak evolution of O2-LLRO is consistent with that of other Ru-based materials during the Li⁺ (de) intercalation process, which is relevant to the connective cationic and anionic redox reactions[45,49,50]. As well, X-ray photoelectron spectroscopy (XPS) was used to study the charge compensation mechanism of

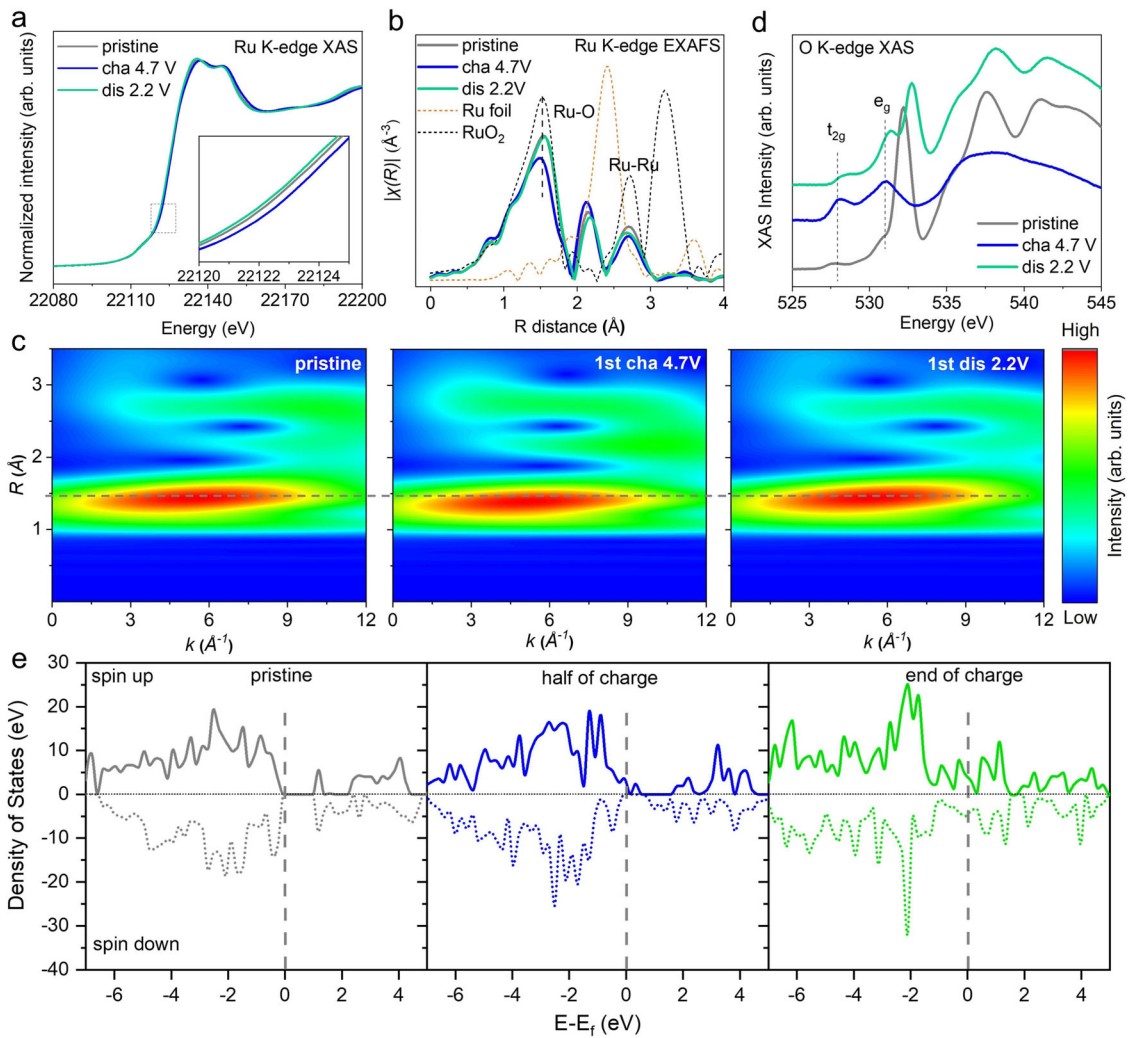

**Fig. 5 | Charge compensation mechanism of LLRO. a** Ru K-edge XANES spectra. (Inset: amplifying area). **b** Ru K-edge EXAFS spectra. **c** Ru K-edge WT EXAFS spectra XAFS images of LLRO. **d** O K-edge sXAS spectra. **e** pDOS at different charge states. Source data are provided as a Source Data file.

Ru redox furtherly (Supplementary Fig. 14). The peak located at 282.1 eV is assigned to Ru[4+], which shifts to higher energy upon charging to 4.7 V, indicating the oxidation of Ru[4+]. After the Li[+] reinsertion process, the peak moves back to the position similar to that of the pristine, showing a reversible redox of Ru.

As for the anionic redox activities of O2-LLRO, O K-edge sXAS shows that there are two peaks at around 528 eV and 531 eV of the pre-edge region, corresponding to the $t_{2g}$ and $e_g$ band regarded as electron transitions from the unoccupied states of O $2p$ orbitals mixed with Ru $4d$ empty orbitals (Fig. 5d). It should be noted that the sharp peak located at 532.5 eV is assigned to the π* orbital in carbonate anions of $Li_2CO_3$ residual on the material surface[51]. Two broad peaks above 535 eV are attributed to the O $2p$ orbitals hybridized with Ru $5s$ and $5p$ states. At the fully charged state, the carbonate signals disappear, and the intensity of O K-edge pre-edge shows an apparent increase. The evolution of the peak intensity is associated with the removal of electrons from oxygen and the increment of the number of holes in the $t_{2g}$ and $e_g$ orbitals, which renders direct evidence of the oxygen redox for charge compensation[52–54]. Then, the signals of carbonate reappear, and the intensity of pre-edge returns close to its original state after the first discharge, manifesting that the oxygen holes are repopulated with the reversible reduction of oxygen, which is the reason for the extra capacity storage ability of Li-rich layered materials. The invertible change of carbonate during the initial cycle accords well with the work reported by Yang et al.[55].

The partial density of state (pDOS) of oxygen was calculated to theoretically confirm the anionic redox activity (Fig. 5e). At the pristine state, O $2p$ energy level does not appear near the Fermi level, illustrating anions have no redox activity at low voltage regions and cations first start to be oxidized during the initial de-lithiation process. Oxygen begins to be triggered to participate in the charge compensation at the half-charged state since O $2p$ energy level appears around the Fermi level. The pDOS of oxygen sharply increases around the Fermi level at the end of the charge, demonstrating that oxygen predominantly contributes to the capacity.

**Oxygen redox behavior**

In this part, the evolution behavior and sluggish kinetics of oxygen in O2-LLRO during the charge/discharge process have been disclosed in detail. Raman test is a powerful tool for detecting oxygen redox due to its sensitivity to the evolution of oxygen behavior (O-O dimer) and has been utilized broadly to monitor the formation of oxygen species in many Li-rich materials[56–58]. In this case, ex situ Raman spectra of LLRO were collected in Fig. 6a, and the results unequivocally validate the reversible evolution of peroxo-species O-O bond at around 860 cm[-1]. Obviously, at the end of the charge, superoxo-species O-O bond was also detected at around 1100 cm[-1] (Supplementary Fig. 15). Note that the peak at ~790 cm[-1] may belong to the $PF_6^-$ in the electrolyte[56]. Besides, operando differential electrochemical mass spectrometry (DEMS) was also conducted to detect the oxygen activities of LLRO

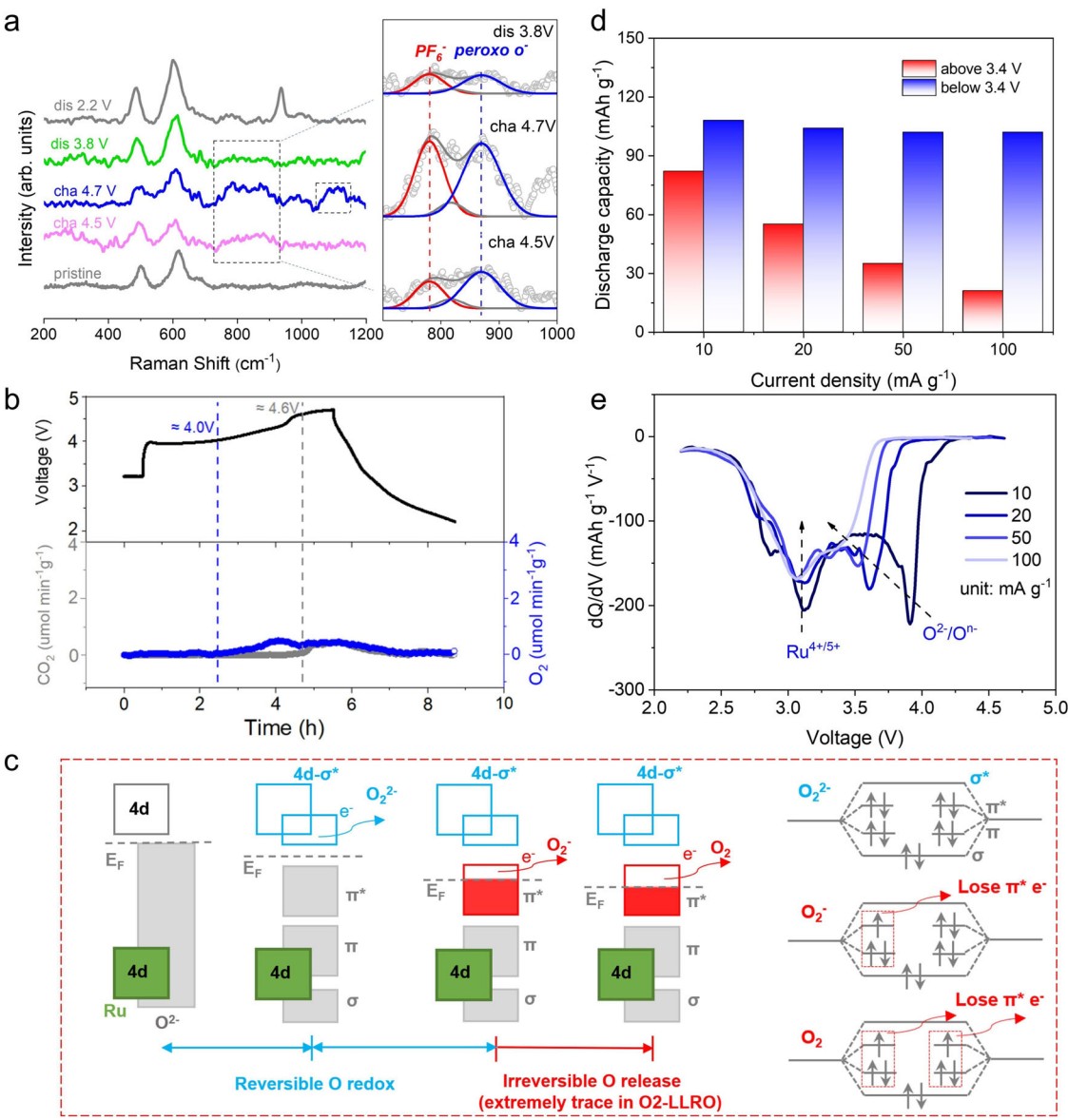

**Fig. 6 | Oxygen redox behavior of LLRO. a** Ex situ Raman spectra of LLRO at the different voltage states. (Inset: enlarged area marked by the dash line at around 800 cm⁻¹). **b** DEMS spectra of LLRO with corresponding charge/discharge curves at 10 mA g⁻¹. **c** Schematic diagrams of the evolution of oxygen band structure and the evolution of electron structure during the initial oxygen oxidation process. **d** Discharge capacity for the two regions (2.2−3.4 V and 3.4−4.7 V) at various current densities. **e** The dQ/dV profiles of the discharge process at different current densities. Source data are provided as a Source Data file.

during the first Li⁺ (de)intercalation (Fig. 6b). Profited from the strong Ru-O covalent bond, trace O₂ loss derived from the over-oxidation of lattice oxygen can be detected (μmol min⁻¹ g⁻¹ level), which is far less than the previous results of O3-Li₂RuO₃ cathode[45,50,52], indicating that more reversible oxygen evolution can be achieved in O2-LLRO. What's more, CO₂ evolution also appeared at the end of the charge caused by the decomposition of electrolyte and/or the decomposition of surface carbonate. In short, by combining Raman and DEMS characterizations, the oxygen evolution process of LLRO could be streamlined step by step (O²⁻ to O⁻ to O₂⁻ to trace O₂) upon the voltage lifting[57,59]. The oxygen evolution process in O2-LLRO is quite different from that of Na₀.₆Li₀.₂Mn₀.₈O₂ with ribbon-ordered superstructure and Na₀.₇₅Li₀.₂₅Mn₀.₇₅O₂ with honeycomb-ordered superstructure reported by House et al. [14], in which the oxygen evolution species of two samples can only be detected as trapped molecular O₂. The discrepancy highlights the importance of pristine structure, which may directly affect the oxygen evolution path.

To receive a deeper understanding of anionic redox, schematic diagrams about the evolution of the oxygen band and electron structure during the initial charge process are shown in Fig. 6c. In detail, when oxygen begins to participate in the charge compensation process, O 2p σ* orbital overlaps with the Ru 4d orbital to lie above the Fermi level and fully filled π*, π, σ orbitals lie below the Fermi level, resulting in the reversible redox actions between lattice O²⁻ and O⁻. Next, the Fermi level decreases constantly upon further charge, and the electrons continuously escape from π* orbital, which is an irreversible redox process, leading to the formation of O₂⁻ and then gaseous O₂ loss. The survived peroxo-species after the full charge could be reduced back to O²⁻ and thus contribute to the discharge capacity in the subsequent Li⁺ reinserted process.

It is commonly considered that oxygen redox activities are sluggish and usually asymmetrical, crossing the whole discharge voltage regions[21,60]. The asymmetrical behavior is related to the variation of the oxygen coordination environment and is directly affected by TM

migration. Benefiting from the reversible OO migration of Ru, the asymmetrical behavior of oxygen redox is effectively restricted in the superstructure-free O2-LLRO, exhibiting a vast difference with the reported disordered O2-LLCM[24]. Supplementary Fig. 16 shows the discharge curves of O2-LLRO at different current densities, in which the capacity variations offered by redox of cationic and anionic are readily recognized. Intuitively, Fig. 6d manifests that the capacity of high potential rapidly fades from 85 mAh g$^{-1}$ at 10 mA g$^{-1}$ to 21 mAh g$^{-1}$ at 100 mA g$^{-1}$, while the redox region below 3.4 V is well sustained with minimal changes of no more than 9 mAh g$^{-1}$ as the current increasing. It is widely believed that the former is rooted in the reduction of oxygen species, and the latter is contributed by the reduction of cations[61]. Indeed, the considerable difference further confirms the sluggish kinetics of oxygen redox at high voltage, which is also supported by the dQ/dV curves (Fig. 6e). However, the peaks attributed to the reduction of Ru$^{4+/5+}$ are always immobilized at ~3.1 V regardless of the variation of current density, indicating a highly reversible Ru migration process and hence a stable oxygen coordination environment. The above results confirm the high oxygen redox symmetry in the superstructure-free O2-LLRO. As a stark contrast, discharge capacity shows a significant decrease in both regions for O3-Li$_2$RuO$_3$ (Supplementary Fig. 17), which is ascribed to oxygen reduction dropping into low voltage region deriving from the irreversible Ru migration upon the increase of current density, suggesting the more severe asymmetry of anionic redox for O3-Li$_2$RuO$_3$.

## Discussion

In this study, we highlight that reversible TM migration can still be achieved in a superstructure-free O2-type oxide. The O2-LLRO cathode was originally designed as the platform to investigate the unexplored mechanism of TM migration and redox reactions in O2-type Ru-based layered oxides. We discover that the migration of Ru in superstructure-free O2-type oxide follows a quite different path from that of the traditional Mn case. Theoretical calculation results show that Ru tends to migrate from O$_{TM}$ to the intermedia O$_{AM}$ instead of T$_{AM}$ first and stay at the intermedia O$_{AM}$ sites finally, which streamlines the return path to the original sites upon the discharge process. As a result, the O2-LLRO presents the highly reversible TM migration, thus leading to robust structural stability and oxygen redox symmetry, which collectively contributes to the excellent capacity (90.2% after 100 cycles) as well as voltage stability (97.7% voltage retention after 100 cycles) of the cathode. This work declares that except the superstructure ordering, more factors such as the covalency with oxygen and atom radius should be considered to achieve highly reversible TM migration with respect to the future design, which provides an instructive insight into designing Li-rich layered oxides with suppressed voltage decay and stable oxygen redox activities.

## Methods

### Synthesis

Na$_{0.6}$Li$_{0.2}$Ru$_{0.8}$O$_2$ material was prepared by traditional solid-state methods from Na$_2$CO$_3$, Li$_2$CO$_3$, and RuO$_2$ of stoichiometric ratio with an excess of 5 mol% Na$_2$CO$_3$, Li$_2$CO$_3$. Three powders were ground thoroughly for half an hour. The obtained powder was pelleted and calcined at 900 °C for 12 h under air. After cooling to room temperature (25 °C), the Na-deficient precursor Na$_{0.6}$Li$_{0.2}$Ru$_{0.8}$O$_2$ was transferred into the glove box for further ion exchange.

Li$_{0.6}$Li$_{0.2}$Ru$_{0.8}$O$_2$ material was prepared by ion exchange method from Na-precursor Na$_{0.6}$Li$_{0.2}$Ru$_{0.8}$O$_2$. Firstly, the Na$_{0.6}$Li$_{0.2}$Ru$_{0.8}$O$_2$ and 10 times Li excess molten salt (eutectic mixture of 88 mol% LiNO$_3$ and 12 mol% LiCl) were mixed for 1 h. Then, the mixture was calcined at 280 °C for 3 h in a muffle furnace. After cooling to room temperature, Li$_{0.6}$Li$_{0.2}$Ru$_{0.8}$O$_2$ can be obtained after washing with distilled water and then dried at 60 °C overnight.

Li$_2$RuO$_3$ material was prepared by traditional solid-state methods from Li$_2$CO$_3$ and RuO$_2$ of stoichiometric ratio with an excess of 5 mol% Li$_2$CO$_3$. The powders were ground thoroughly for half an hour. The obtained powder was pelleted and calcined at 950 °C for 15 h under air. After cooling to room temperature, Li$_2$RuO$_3$ was transferred into the glove box for further electrode preparation.

### Electrochemical tests

2032 coin-type cells were used for electrochemical measurements and were assembled in an Ar-filled glove box. Electrodes were prepared with active material, Super P, and polytetrafluoroethene (12 wt.%) binder with a weight ratio of 70:20:10. The diameter, area, and mass loading of each electrode is about 6 mm, 0.3 cm$^2$, and 3 mg cm$^{-2}$, respectively. Al mesh is used as the current collector. 1 M LiPF$_6$ in ethylene carbonate and diethyl carbonate (1: 1 in volume) was used as the electrolyte, and a glass fiber film was employed as a separator. The galvanostatic charge-discharge tests of different conditions were performed by using a Land battery testing system (Wuhan, China) at 25 °C. GITT test was performed at the current density of 10 mA g$^{-1}$ and tested 1 h with current flux followed by 5 h rest within the voltage window of 2.2–4.7 V.

### Structural characterizations

The XRD patterns of samples were identified by a Rigaku SmartLabSE X-ray diffractometer (Cu source, λ = 1.54056 Å) in the 2θ range of 10°−80°. In situ XRD patterns were tested from 10°−60° with 26 min and an interval of 4 min; The morphology of the powder was characterized by field emission scanning electron microscope (FE-SEM, Carl Zeiss Sigma 500); HAADF-STEM, ABF-STEM, and IDPC-STEM image were collected at different cycled states and performed at Themis Z (3.2) at Thermo scientific with focused ion beam cut technology.

### Ex situ measurements

Ru K-edge hard X-ray absorption spectra (XAS) and O K-edge soft X-ray absorption spectra (sXAS) were collected at spring8 BL14B2 by Japan Synchrotron. All samples were prepared as films with active material of 25 mg; XPS was characterized by using Thermo Scientific K-Alpha, samples were collected at different cycling states and washed by DME before testing; Operando DEMS measurements were conducted by Hiden HPR40 at the current density of 10 mA g$^{-1}$; Raman spectroscopy was performed on the Lab RAM HR Evolution Laser Raman spectrometer.

### Solid-state $^7$Li nuclear magnetic resonance

Bruker-AVANCE-500M NMR spectrometer (11.74 T) with the $^7$Li Larmor frequency of 194.3 MHz using a 1.3 mm double-resonance MAS probe was used to perform the $^7$Li magic angle spinning (MAS) NMR experiments. 1.3 mm MAS rotors were filled by the samples in an argon-filled glove box and rotated at the spinning rate of 55 kHz. To eliminate the sidebands, the pj-MATPASS experiment was used, in which π/2 pulse width was 1.5 μs and a recycle delay was 0.1 s; 1 M LiCl was used as the reference of 0 ppm chemical shift. For each set of measurements, we prepared 4 electrodes with mass loading of around 6.5 mg cm$^{-2}$. After cycling, we disassembled the cells in an Ar-filled glove box and washed the electrodes with electrolyte solvent. We cut the electrodes into pieces and then sealed them in a centrifuge tube for the NMR measurements.

### Theoretical calculations

The first-principles simulations based on density functional theory (DFT)[62,63] were implemented in Vienna ab initio Simulation Package (VASP 5.4.4 version[64]) with the projector augmented wave pseudopotential method[65,66]. The generalized gradient approximation (GGA) with the Perdew-Burke-Ernzerhof[67] version was used for calculations[68,69]. The supercells were used to investigate the Ru

migrations, which contain 66 atoms for $Li_{0.67}(Li_{0.22}Ru_{0.78})O_2$ and 60 atoms for $Li_{0.33}(Li_{0.22}Ru_{0.78})O_2$ with the U parameters of 4.0 eV for Ru. A plane-wave energy cutoff of 520 eV was chosen for structure relaxation and DOS calculations. The Gamma-centered $2 \times 2 \times 2$ k-mesh was applied for the Brillouin zone sampling. The convergence criteria of structural optimization were $10^{-5}$ eV/atom for total energy and 0.01 eV $\text{Å}^{-1}$ for forces. The optimized structure of LLRO is shown in Supplementary Fig. 7, where we performed $3 \times 3 \times 1$ cell expansion of the structure to get all the possible structures under the $Li_{16}Ru_{14}O_{36}$ composition, arranged according to Ewald energy from smallest to largest and calculated the DFT structure energy to get the most stable structure.

## Reporting summary

Further information on research design is available in the Nature Portfolio Reporting Summary linked to this article.

## Data availability

All other relevant data are available from the corresponding author upon reasonable request. Source data are provided with this paper.

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

## Acknowledgements

X.L. acknowledges the support of the China Postdoctoral Science Foundation (grant nos. 2023T160591). Y.F. acknowledges the support of the National Natural Science Foundation of China (grant nos. 22350710185, 22005274). Y.X. acknowledges the support of the Westlake Education Foundation.

## Author contributions

T.C. and J.X. contributed equally to this work. T.C. designed and carried out the experiments, analyzed data, and wrote the manuscript. X.L. and Y.F. directed the research and supervised the manuscript. J.X. and H.Z. performed the theoretical calculations. X.W. helped analyze the STEM results. L.L. analyzed the XAS results. Y.X. performed the $^7$Li solid-state NMR. All authors have given approval to the final version of the manuscript.

## Competing interests
