## [Peer Review File · Nature Communications]

Highly reversible transition metal migration in superstructure-free Li-rich oxide boosting voltage stability and redox symmetryREVIEWER COMMENTS

Reviewer #1 (Remarks to the Author):

This manuscript reports the structural and chemical evolution of a O₂-type Li-rich Ru oxide layered cathode. The authors found good electrochemical properties of this material, compared with other Ru based compounds like Li₂RuO₃, in terms of cycling stability and voltage retention. The authors employed electron microscope and various spectroscopy techniques to verify the cycling mechanism and concluded that the improved reversibility is from the specific OO Ru migration that is different from Li₂RuO₃. The authors then further claim the independence of superstructure for reversible Ru migration as an impactful finding for material explorations.

As a reviewer, I like the choice of the system and appreciate the efforts of the authors in the synthesis and analysis of such an intriguing system. The electrochemical performance, especially in capacity and voltage retention, is obviously better than Li₂RuO₃, although with a relatively lower capacity for the obvious reason. These findings, especially the different Ru migration channel, are interesting to me in general, but I'm yet to be convinced on the final claim, as well as some technical comments:

Main concern:

I would like to note first that the heavily cited Ref. 14 actually emphasizes two things: the "lost" honeycomb superstructure of Na_{0.75}Li_{0.25}Mn_{0.75}O₂ upon charging, and the structural dependence of O₂ molecule based oxygen redox mechanism by considering the space needed to form the molecule inside the electrodes.

The material studied in this work does show no signature of superstructure in its pristine state, but the honeycomb ordering is pretty clear at the charged state as discussed in Fig. 3. So, the interesting contrast here is not really the "superstructure-free" as claimed many times in the paper; instead, the superstructure exists at least during the important initial cycle but behaves in a very different way from typical Li-rich compounds. Most oxygen redox systems show different kind of superstructures, but they also display changes in its superstructure upon electrochemical cycling. Many Li-rich compounds, e.g., the most common Li_{1.2}NCM compounds, loses superstructure after the conditioning process but maintain very good reversibility in later cycles. So I don't understand why the authors keep emphasizing this material is "superstructure-free", to me, it is the behavior of the superstructure in this material that is different from others, instead of free from superstructure.

Another important difference is that Ref. 14 raised the model of oxygen redox based on O₂ molecule formation. All the superstructural discussions there were actually towards the O₂ molecule formation. But what the authors found here is completely different. The Raman spectra clearly show the oxidized oxygen species here are peroxy or superoxol, instead of trapped O₂ that is at very different wavenumber. I am very surprised that such a huge difference is completely ignored in the oxygen redox mechanism discussion because this indicates a completely different redox mechanism in a Li-rich compound.

I should clarify that I'm not questioning the authors findings, but just feel the discussions and conclusions are away from what their data say.

Other technical comments:

- It's hard to understand the discussions of oxygen redox based on the absorption and XPS analysis. The O K-edge absorption peaks are strongly affected by the Ru orbitals as the authors correctly pointed out, but the main change there seems to be the carbonate formation and decomposition instead of indicating any oxygen redox signatures. Such surface carbonate evolution is also clear in the XPS data. I also cannot agree the claim that

“The O K-edge absorption spectrum of the fully charged state is extremely similar to that of other Li-rich materials involving anionic redox” – they look very different in the number of peaks and the location of energies, at least not “extremely similar”.

- Multiple groups have reported that absorption and XPS techniques are incapable of concluding oxygen redox behaviors in battery cathodes. But Li_2RuO_3 has been thoroughly studied by RIXS with clear evidence of Ru and oxygen redox. So I think the authors statement on the general redox mechanism should be correct although I would suggest the authors consult with someone familiar with spectroscopy to clarify their discussions. To me, the Raman data is the most conclusive evidence for discussing the oxygen redox in this work.

- I don't see the meaning of Fig. 6c and the whole paragraph of description in the text. This seems to be just a general schematic of a very vague model that is popular, but not necessarily true, in the battery field. What is the useful information on this particular material or the relationship with the interesting Ru migration here?

- The authors consider CO_2 “caused by the decomposition of electrolyte or the nucleophilic attack by superoxo O_2^- ”. But their absorption data already show clearly that it is the surface carbonate formation during discharging and decomposition during charging. The effect seems to dominate their spectral change (Fig. 4a and 5d) and could easily explain the CO_2 release.

Reviewer #2 (Remarks to the Author):

In this manuscript, the authors report a superstructure-free O_2 -type oxide, which shows decent cyclability and minimal voltage decay. With the help of structural & chemical characterization and theoretical calculation, they propose that the improved structural stability and oxygen redox symmetry is a result of reversible transition metal migration. The most noteworthy result from this work is that it demonstrates the possibility of mitigating irreversible TM migration without the help of superstructure ordering, which provides new insights for Li-rich cathode material design.

In general, this work contains novel findings that are well supported by its logic flow and datasets. The experiments are well designed, and detailed methodology are provided for reproduction. However, there are still some flows need to be revised to clearly illustrate the key points of the paper. Thus, I recommend the manuscript to be accepted for publication only after the comments below are well addressed.

1. In Figure 3a, 3b, the authors use HADDF-STEM to demonstrate the reversibility of Ru migration during charge/discharge, but only analysis on 1st cycle has been done. Although Figure 4e shows ex-situ XRD pattern of cycled cathode, I still think HADDF-STEM results on cycled O_2 -LLRO material is necessary to verify the structural stability of such material during cycling.

2. According to Figure 2a and comparing Figure 1c with Figure 3b, there is an obvious irreversible structural change during first cycle. The authors should better explain such 1st

cycle irreversibility.

3. In Line 373-375b and Figure 6d & Figure S15, authors use capacity retention below 3.4V at increased current density to demonstrate stable Ru redox. This is flawed, since the overpotential is not constant under different current densities and for constant current discharge at 4.3V, the equilibrium potentials are very different under different current rates.

4. In Figure S13b and Line 325-327, authors claim that 'When charged to 4.7 V, a new peak located at 530.5 eV emerges, which is attributed to the formation of oxidized O_n⁻, confirming that the 326 anionic oxidation is triggered during the first charge process'. However, to me there is no obvious distinction between O1s spectra of LLRO at 2.2V and at 4.7V. Authors should either clearly explain their XPS fitting method or delete such argument.

5. I noticed that for electrochemical test in this work, 1 M LiPF₆ in ethylene carbonate (EC) and Diethyl carbonate (DEC) electrolyte was used. It is known that such type of carbonate electrolyte tends to get oxidized at >4.5V, but the cells in this work are cycled between 2.2V-4.7V. Please analyze the possibility of electrolyte side-reactions and how does it affect the analysis.

Reviewer #3 (Remarks to the Author):

In this study, the authors highlight a reversible transition metal migration phenomenon even in a superstructure-free O₂-type oxide, which is different from the typical Mn-based O₂-type oxide. They uncover that the migration of Ru in superstructure-free O₂-type oxide follows a quite different path from that of traditional Mn case by HAADF-STEM and theoretical calculations, which provides an instructive insight into understanding and designing Li-rich layered oxides with suppressed voltage decay. The experimental and calculation data is complete, which bears strongly convincing. The work is innovative and crucial to promote the development of the field, and I suggest that this work be published on Nature communications after a proper revision concerning these comments.

1. What is the principle of synthesizing materials by ion exchange method? What factors should be paid special attention to during the ion exchange process? The discussion should be added, which is useful for the readers to better comprehend the novelty of this method.

2. Why does the author design the O₂-type material as Li_{0.6}Li_{0.2}Ru_{0.8}O₂? Can we obtain an O₂-type material with a high Li-ion concentration?

3. As the authors evidenced, O₂-type Mn-based Li-rich cathodes show a large difference with Ru-based Li-rich cathodes. Can the authors give a comparison between them, about the influencing factors such as superstructure ordering, TM migration, and other potential factors affecting the electrochemical behavior?

4. How to understand the pattern in Figure 1f marked as "stacking faults"? Do the authors think the stacking faults are important to the electrochemical behavior?

5. The O₂-type Li_{0.6}Li_{0.2}Ru_{0.8}O₂ and O₃-type Li₂RuO₃ exhibit different discharge behavior, as shown in Figure 6d and Figure S16. How to understand the phenomenon?

NCOMMS-23-61143
RESPONSE LETTER TO REVIEWERS

Reviewer #1:

This manuscript reports the structural and chemical evolution of a O₂-type Li-rich Ru oxide layered cathode. The authors found good electrochemical properties of this material, compared with other Ru based compounds like Li₂RuO₃, in terms of cycling stability and voltage retention. The authors employed electron microscope and various spectroscopy techniques to verify the cycling mechanism and concluded that the improved reversibility is from the specific OO Ru migration that is different from Li₂RuO₃. The authors then further claim the independence of superstructure for reversible Ru migration as an impactful finding for material explorations.

As a reviewer, I like the choice of the system and appreciate the efforts of the authors in the synthesis and analysis of such an intriguing system. The electrochemical performance, especially in capacity and voltage retention, is obviously better than Li₂RuO₃, although with a relatively lower capacity for the obvious reason. These findings, especially the different Ru migration channel, are interesting to me in general, but I'm yet to be convinced on the final claim, as well as some technical comments:

Reply: We thank you for your positive comments sincerely.

Main concern:

1. I would like to note first that the heavily cited Ref. 14 actually emphasizes two things: the “lost” honeycomb superstructure of Na_{0.75}Li_{0.25}Mn_{0.75}O₂ upon charging, and the structural dependence of O₂ molecule based oxygen redox mechanism by considering the space needed to form the molecule inside the electrodes.

The material studied in this work does show no signature of superstructure in its pristine state, but the honeycomb ordering is pretty clear at the charged state as discussed in Fig. 3. So, the interesting contrast here is not really the “superstructure-free” as claimed many times in the paper; instead, the superstructure exists at least during the important initial cycle but behaves in a very different way from typical Li-rich compounds. Most oxygen redox systems show different kind of superstructures, but they also display changes in its superstructure upon electrochemical cycling. Many Li-rich compounds, e.g., the most common Li_{1.2}NCM compounds, loses superstructure after the conditioning process but maintain very good reversibility in later cycles. So I don't understand why the authors keep emphasizing this material is “superstructure-free”, to me, it is the behavior of the superstructure in this material that is different from others, instead of free from superstructure.

Reply:

We are very thankful for your good advice. We feel sorry to cause misunderstanding for you because of “superstructure-free”.

We think the state of the superstructure plays a key effect on the initial electrochemical performance. Kisuk Kang et al. found that superstructure ordering has an effect on TM migration and oxygen redox symmetry of O₂-type layered oxides.¹ For

better comparison, they obtained two samples of O2-type $\text{Li}(\text{Li}_{0.25}\text{Ni}_{0.125}\text{Mn}_{0.625})\text{O}_2$ (LLNM) with superstructure and O2-type $\text{Li}(\text{Li}_{0.25}\text{Co}_{0.25}\text{Mn}_{0.5})\text{O}_2$ (LLCM) without superstructure (superstructure-free), which was differentiated based on their pristine structure. They revealed that the former exhibited more reversible TM migration and symmetrical oxygen redox than the latter. Therefore, they highlight the significance of the initial superstructure for achieving reversible oxygen redox in that paper. However, they didn't show if there exists a superstructure in LLCM material during the initial charge process.

Herein, it was our initial belief that if there are some other crucial factors besides initial superstructure to affect the reversibility of TM migration and oxygen redox symmetry in O2-type layered oxides. The size of TM was the first point we considered. Therefore, we determined to design a Ru-based O2-type material without a pristine superstructure. Indeed, the superstructure would form in our LLRO after the initial charge process, but it disappeared after the first cycle and did not occur after the second charge process (see Figures 3b and 3i in the manuscript). More importantly, the superstructure-free was found to be maintained even after 10 cycles (**Figure R1**). In fact, the superstructure after the initial charge process appeared by chance due to the absence of Li ions in the TM layer as we demonstrated in the manuscript (relative description of Figure 4a), but it was not the persistent presence during the long-term cyclic process. Therefore, herein we modestly regarded LLRO as a superstructure-free material. Anyway, we successfully proved that the superstructure ordering (at least for the initial state) is not necessary to achieve reversible TM migration and symmetrical oxygen redox in O2-type layered oxides, providing a new insight into designing stable Li-rich cathodes with oxygen redox. To prevent misunderstandings, we have changed “superstructure-free” to “pristine superstructure-free” in the title of our revised version.

We hope our explanation meets your satisfaction.

Figure R1. HAADF-STEM of O2-LLRO after 10 cycles.

2. Another important difference is that Ref. 14 raised the model of oxygen redox based on O₂ molecule formation. All the superstructural discussions there were actually towards the O₂ molecule formation. But what the authors found here is completely different. The Raman spectra clearly show the oxidized oxygen species here are peroxo or superoxol, instead of trapped O₂ that is at very different wavenumber. I am very surprised that such a huge difference is completely ignored in the oxygen redox mechanism discussion because this indicates a completely different redox mechanism in a Li-rich compound. I should clarify that I'm not questioning the authors findings, but just feel the discussions and conclusions are away from what their data say.

Reply:

We thank you for your careful research.

Indeed, according to House's findings in Ref.14, O₂ molecule will form at the end of the charge process and be trapped in the bulk, meaning that O₂ molecule may well be one of the final products of oxygen oxidation. However, the possibilities are not the only one. Importantly, Tarascon et al. proposed a reductive coupling mechanism by DFT to stabilize the highly oxidized Ru⁶⁺ cation by modifying its coordination sphere into either Ru⁵⁺-(O₂)²⁻ or Ru⁴⁺-(O₂)⁻ in Li₂Ru_{0.75}Sn_{0.25}O₃ system (**Figure R2a**),² showing that O₂²⁻ and O₂⁻ are rather important during the oxygen redox process, which was further verified by Electron Paramagnetic Resonance (EPR) (**Figure R2b**). That is to say, they successfully detected superoxol O₂⁻ (the signal of O₂²⁻ is EPR-silent).³ Besides, with the aid of STEM and neutron diffraction, they also demonstrated that peroxol O₂²⁻ would be formed at the end of the charge process in Li₂IrO₃ system (**Figure R2c**).⁴ It should be noted that Li₂RuO₃ and Li₂IrO₃ systems both exhibit honeycomb superstructure. What's more, our previous works also detected O₂²⁻ and O₂⁻ by Raman in several Li⁺/Na⁺ cathode materials through operando test.⁵⁻⁷

As mentioned above, oxygen ions may evolve along the way of O²⁻-O₂²⁻-O₂⁻-O₂, instead of direct formation from O²⁻ to O₂. Although O₂ molecules in the **bulk** may be trapped upon the charge process and be reduced during the discharge process, the ones at the **surface** will be released and detected by DEMS.

Figure R2. a) Reductive coupling mechanism and calculations for peroxo-like species.² **b)** EPR results of $Li_2Ru_{0.75}Sn_{0.25}O_3$.³ **c)** Fraction of lattice oxygen attributed to peroxo for $Li_yIr_{1-x}Sn_xO_3$ samples (purple line: $x=0$, orange line: $x=0.25$).⁴

Other technical comments:

3. It's hard to understand the discussions of oxygen redox based on the absorption and XPS analysis. The O K-edge absorption peaks are strongly affected by the Ru orbitals as the authors correctly pointed out, but the main change there seems to be the carbonate formation and decomposition instead of indicating any oxygen redox signatures. Such surface carbonate evolution is also clear in the XPS data. I also cannot agree the claim that "The O K-edge absorption spectrum of the fully charged state is extremely similar to that of other Li-rich materials involving anionic redox". They look very different in the number of peaks and the location of energies, at least not "extremely similar".

Multiple groups have reported that absorption and XPS techniques are incapable of concluding oxygen redox behaviors in battery cathodes. But Li_2RuO_3 has been thoroughly studied by RIXS with clear evidence of Ru and oxygen redox. So I think the authors statement on the general redox mechanism should be correct although I would suggest the authors consult with someone familiar with spectroscopy to clarify their discussions. To me, the Raman data is the most conclusive evidence for discussing the oxygen redox in this work.

Reply:

We appreciate your concerns about the manuscript sincerely.

Indeed, it is hard to get more information from O K-edge absorption caused by the strong disturbance by Ru orbitals. And XPS is also not suitable for exploring oxygen redox because several oxygen species strongly overlap with others. To keep the research

rigorous, therefore, we also employed Raman and DEMS to explore the oxygen redox behaviors in the manuscript. According to the results of Raman and DEMS, we further confirmed the participation of oxygen redox during cycling. What's more, according to your advice, we have deleted the misleading description: "The O K-edge absorption spectrum of the fully charged state is extremely similar to that of other Li-rich materials involving anionic redox".

Moreover, we have consulted with someone familiar with XAS and compared with relative Li_2RuO_3 system according to your advice.^{8,9} Since e_g orbital in our LLRO is significantly affected by carbonate signals, we mainly focus on the variation of t_{2g} orbital. In the previous reports,^{8,9} we find that although the oxidation of Ru would affect the t_{2g} orbital, the variation caused by which is very small and t_{2g} orbital becomes broad and round (see the t_{2g} variation at the low voltage region in **Figure R3a** and 3b). Once the electrode is charged to more than 4.3V, t_{2g} orbital becomes relatively sharp, which is very similar to the 4.7 V-charged state of our LLRO (Figure 5d). We know that there is a little one-sided by comparing the peak shape of t_{2g} orbital, but it still bears a certain persuasive force. Furthermore, we want to emphasize that the results of sXAS and XPS may contain misleading information, while Raman will not, which can give direct and solid evidence about oxygen evolution.

We feel sorry again for the misleading and hope our explanation could meet your satisfaction.

Figure R3. O K-edge of O k-edge of a) ID- Li_2RuO_3 and b) $\text{Li}_2\text{Ru}_{0.75}\text{Fe}_{0.25}\text{O}_3$.^{8,9}

4. I don't see the meaning of Fig. 6c and the whole paragraph of description in the text. This seems to be just a general schematic of a very vague model that is popular, but not necessarily true, in the battery field. What is the useful information on this particular material or the relationship with the interesting Ru migration here?

Reply:

We feel sorry to cause some concern for you.

In Figure 6c of the manuscript, the description is used to clear the evolution process of oxygen participating in charge compensation. With the aid of XAS, XPS, Raman, DEMS, we propose that oxygen may evolve along the path of O^{2-} - O_2^{2-} - O_2^- - O_2 . It is our belief that a schematic diagram should be drawn to summarize the above characterizations associated with oxygen redox, which will be convenient for the readers.

Indeed, the key point of this paper is TM migration. However, O2-LLRO is a novel cathode that has not been reported before. Therefore, we modestly believe that besides structural evolution, the inherent charge compensation mechanism of this material during cycling should also be shed light on. Meanwhile, Figure 6c could also introduce the following exploration of oxygen redox symmetry, serving as a link between the above and the below.

We feel sorry again for the misleading and hope our explanation can meet your satisfaction.

5. The authors consider CO_2 “caused by the decomposition of electrolyte or the nucleophilic attack by superoxo O_2^- ”. But their absorption data already show clearly that it is the surface carbonate formation during discharging and decomposition during charging. The effect seems to dominate their spectral change (Fig. 4a and 5d) and could easily explain the CO_2 release.

Reply:

Thank you for your advice. As you said, the electrolyte would decompose at high voltages.^{10, 11} Some carbonate species can be formed by the parasitic reaction from the electrolyte. Combining your comments, we have updated the relative description as “ CO_2 evolution also appeared at the end of the charge **caused by the decomposition of electrolyte and/or the decomposition of surface carbonate**”.

We hope that our responses meet your satisfaction and the manuscript is now suitable to be accepted by *Nature Communications*.

Reviewer #2:

In this manuscript, the authors report a superstructure-free O2-type oxide, which shows decent cyclability and minimal voltage decay. With the help of structural & chemical characterization and theoretical calculation, they propose that the improved structural stability and oxygen redox symmetry is a result of reversible transition metal migration. The most noteworthy result from this work is that it demonstrates the possibility of mitigating irreversible TM migration without the help of superstructure ordering, which provides new insights for Li-rich cathode material design.

In general, this work contains novel findings that are well supported by its logic flow and datasets. The experiments are well designed, and detailed methodology are provided for reproduction. However, there are still some flows need to be revised to clearly illustrate the key points of the paper. Thus, I recommend the manuscript to be accepted for publication only after the comments below are well addressed.

Reply: We thank you for your positive comments sincerely.

1. In Figure 3a, 3b, the authors use HAADF-STEM to demonstrate the reversibility of Ru migration during charge/discharge, but only analysis on 1st cycle has been done. Although Figure 4e shows ex-situ XRD pattern of cycled cathode, I still think HAADF-STEM results on cycled O2-LLRO material is necessary to verify the structural stability of such material during cycling.

Reply:

Thank you for your useful advice to improve the manuscript. According to your advice, we have performed HAADF-STEM of O2-LLRO after 10 cycles. As shown in **Figure R4**, there are almost no TM ions left in the Li layer, showing a highly reversible TM migration phenomenon in this O2-LLRO cathode, which is consistent with our proposition.

We have added **Figure R4** as **Figure S9** in our revised manuscript: “To further verify the reversibility of TM migration, the HAADF-STEM of O2-LLRO after 10 cycles was also performed. As shown in **Figure S9**, there are almost no TM ions left in the Li layer, showing a highly reversible TM migration of O2-LLRO, which provides a solid structural foundation for outstanding capacity and voltage retention.”

Figure R4. HAADF-STEM of O2-LLRO after 10 cycles.

2. According to Figure 2a and comparing Figure 1c with Figure 3b, there is an obvious irreversible structural change during first cycle. The authors should better explain such 1st cycle irreversibility.

Reply:

For most typical Li-rich layered oxides, the second cycle will always be different from the first cycle, such as $\text{Li}_{1.2}\text{Ni}_{0.13}\text{Mn}_{0.54}\text{Co}_{0.13}\text{O}_2$, $\text{Li}_{1.2}\text{Ni}_{0.2}\text{Mn}_{0.6}\text{O}_2$ and so on (Figure R5),^{2, 7, 12, 13} which must result from the structural rearrangement with irreversible Li ions (in the TM layer) and transition metal migration. Similarly, in our LLRO electrode, the change still occurs during the first cycle. However, it has been demonstrated that the Ru ions migration is highly reversible according to the HAADF-STEM and DFT results. Therefore, attention should be paid to Li ions. As shown in Figure 4a, ⁷Li ss-NMR results show Li_{TM} ions would migrate from the TM layer into the Li layer upon initial charge process, together with migrated Ru ions to leave some vacancies in the TM layer. However, upon the Li ions re-insertion process, nearly all the Li ions would stay in the Li layer because the Li layer is enough to accommodate a total of 0.9 mol Li ions (no more than 1 mol), so there is almost no Li-ion within TM layer after 1st cycle. As a result, the structure after 1st cycle is significantly different from the initial state, leading to different charge/discharge curves.

Figure R5. Initial two cycles curves of a) $\text{Li}_{1.2}\text{Ni}_{0.13}\text{Mn}_{0.54}\text{Co}_{0.13}\text{O}_2$ and b) $\text{Li}_{1.2}\text{Ni}_{0.2}\text{Mn}_{0.6}\text{O}_2$.^{7, 13}

3. In Line 373-375b and Figure 6d & Figure S15, authors use capacity retention below 3.4 V at increased current density to demonstrate stable Ru redox. This is flawed, since the overpotential is not constant under different current densities and for constant current discharge at 4.3 V, the equilibrium potentials are very different under different current rates.

Reply:

We thank you for your advice to improve the manuscript.

The comparison between Figure 6d and Figure S15 is not to demonstrate stable Ru redox, but to say the more asymmetrical oxygen redox in $\text{O3-Li}_2\text{RuO}_3$ than O2-LLRO . As shown in Figure 6e, the peaks corresponding to oxygen reduction are always higher than 3.4 V even though the current density is increasing constantly, which is why

we consider 3.4 V as the critical point. Upon discharge process, in general, the high voltage region corresponds to the reduction of oxygen anions, while the low voltage region corresponds to the reduction of Ru ions. According to the previous reports, the oxygen redox exhibits more sluggish kinetics than cations redox.^{12, 14} Therefore, if the current density increases, there will be an obvious drop of the discharge capacity above 3.4 V for either O2-LLRO or O3-Li₂RuO₃; generally, the discharge capacity below 3.4 V would keep almost unchanged because the kinetics of cations redox is fast. However, for O3-Li₂RuO₃ (Figure S15), upon the current density increases, there is a significant decrease in both areas because the sluggish oxygen redox spreads into the low voltage region, meaning the asymmetry of anionic redox initiated by irreversible TM migration in O3-Li₂RuO₃. While there is no obvious decrease for the discharge capacity below 3.4 V in O2-LLRO (Figure 6d), meaning more asymmetry of anionic redox, which results from the reversible TM migration.

4. In Figure S13b and Line 325-327, authors claim that ‘When charged to 4.7 V, a new peak located at 530.5 eV emerges, which is attributed to the formation of oxidized Oⁿ⁻, confirming that the 326 anionic oxidation is triggered during the first charge process’. However, to me there is no obvious distinction between O1s spectra of LLRO at 2.2 V and at 4.7 V. Authors should either clearly explain their XPS fitting method or delete such argument.

Reply:

We are very thankful for your advice and have deleted the relative descriptions and figures according to your advice. “As well, X-ray photoelectron spectroscopy (XPS) was used to study the charge compensation mechanism of Ru redox furtherly (Figure S13). The peak located at 282.1 eV is assigned to Ru⁴⁺, which shifts to higher energy upon charging to 4.7 V, indicating the oxidation of Ru⁴⁺. After the Li⁺ reinsertion process, the peak moves back to the position similar to that of the pristine, showing a reversible redox of Ru. ~~As for O 1s spectra, there are three peaks of pristine LLRO at 529.1, 531.0, and 532.9 eV vested in lattice O²⁻, C=O, and C-O bond from surface oxygen species, respectively. When charged to 4.7 V, a new peak located at 530.5 eV emerges, which is attributed to the formation of oxidized Oⁿ⁻, confirming that the anionic oxidation is triggered during the first charge process. In addition, the new peak vanishes after discharged to 2.2 V, meaning the invertible evolution of oxygen. The behavior of oxygen redox will be further discussed in the next part.~~

5. I noticed that for electrochemical test in this work, 1 M LiPF₆ in ethylene carbonate (EC) and Diethyl carbonate (DEC) electrolyte was used. It is known that such type of carbonate electrolyte tends to get oxidized at >4.5V, but the cells in this work are cycled between 2.2 V-4.7 V. Please analyze the possibility of electrolyte side-reactions and how does it affect the analysis.

Reply:

As you said, the typical electrolyte cannot endure such high voltages. Indeed, according to the DEMS result in Figure 6b, we detected the formation of CO₂, which is attributed to the decomposition of the ester electrolyte,¹⁰ providing solid evidence of

electrolyte decomposition. Previous reports suggested that EC would be oxidized into Li_2CO_3 and gaseous C_2H_4 (**Figure R6**).¹⁵ The former has been detected by Raman in Figure S15 of the manuscript, which overlaps with the signal of superoxol O_2^- .

As well known, the decomposition of the electrolyte contributes to the formation of cathode electrolyte interphase (CEI). Fortunately, this CEI isolates the contact between the electrolyte and the electrode, preventing the electrolyte from continuous decomposition. Note that the CEI is the product on the interface of the electrode. Our work focuses on the TM migration, which belongs to the bulk information. Therefore, the electrolyte side reactions would not affect our analysis of TM migration.

Figure R6. The equation of EC decomposition.¹⁵

We hope that our responses meet your satisfaction and the manuscript is now suitable to be accepted by *Nature Communications*.

Reviewer #3:

In this study, the authors highlight a reversible transition metal migration phenomenon even in a superstructure-free O2-type oxide, which is different from the typical Mn-based O2-type oxide. They uncover that the migration of Ru in superstructure-free O2-type oxide follows a quite different path from that of traditional Mn case by HAADF-STEM and theoretical calculations, which provides an instructive insight into understanding and designing Li-rich layered oxides with suppressed voltage decay. The experimental and calculation data is complete, which bears strongly convincing. The work is innovative and crucial to promote the development of the field, and I suggest that this work be published on Nature communications after a proper revision concerning these comments.

Reply: We thank you for your positive comments sincerely.

1. What is the principle of synthesizing materials by ion exchange method? What factors should be paid special attention to during the ion exchange process? The discussion should be added, which is useful for the readers to better comprehend the novelty of this method.

Reply:

According to previous reports,¹⁶ the O3-type layered Li-based compounds only can be derived from P3-type and O3-type layered Na precursors, whereas P2-type layered Na precursors can transfer to O2-type, T2-type, and O6-type Li-based layered oxides. Therefore, if the target material is an O2-type layered Li-based compound, P2-type layered Na-based raw material should be obtained.

The driving force during Li^+/Na^+ exchange process is generally deemed the concentration difference or/and temperature. Therefore, the first factor that should be paid attention to is the concentration of molten salt. Generally, more than 10 times the amount of the mixture of 88 mol% LiNO_3 and 12 mol% LiCl is necessary to exchange all Na^+ by Li^+ as far as possible. Second, the factor that should be noted is the temperature. In general, the increased reaction temperature can promote the kinetics of the cation exchange reaction. However, when the temperature is too high, it not only facilitates the diffusion of cations but also enhances the diffusion rate of anions, leading to the instability of the anion sublattice. As such, the ion exchange reaction at high temperature can result in a rapid collapse of the structure. Therefore, the proper temperature is 280°C.

2. Why does the author design the O2-type material as $\text{Li}_{0.6}\text{Li}_{0.2}\text{Ru}_{0.8}\text{O}_2$? Can we obtain an O2-type material with a high Li-ion concentration?

Reply:

There are two principles should be considered. First, to obtain an O2-type Li-rich layered oxide by ion exchange from a Na-based one, the Na-based raw material should be a P2-type layered oxide according to the above discussion. So, we take $\text{P2-Na}_{0.6}\text{Li}_{0.2}\text{Ru}_{0.8}\text{O}_2$ as raw material in the manuscript. Generally, for layered oxides Na_xMO_2 (M= metal), if $0 < x \leq 2/3$, the materials tend to form P2-type; while if $2/3 <$

x , the materials tend to form O3-type. Second, the charge between anions and cations should keep conservation. From these respects, the concentration in the Li layer and transition metal layer could be elevated to $2/3$ and 0.22 , respectively, giving the chemical formula $\text{Li}_{2/3}\text{Li}_{0.22}\text{Ru}_{0.78}\text{O}_2$.

3. As the authors evidenced, O2-type Mn-based Li-rich cathodes show a large difference with Ru-based Li-rich cathodes. Can the authors give a comparison between them, about the influencing factors such as superstructure ordering, TM migration, and other potential factors affecting the electrochemical behavior?

Reply:

According to the report of Kisuk Kang,¹ superstructure-free O2- $\text{Li}_x\text{Li}_{0.25}\text{Co}_{0.25}\text{Mn}_{0.5}\text{O}_2$ (O2-LLCO) shows irreversible transition metal migration. They revealed that there are three migration paths for O2-LLCO (**Figure R7a**): Path I involves the final migration site of the transition metal, which directly neighbors (face-sharing) with other transition metals in the layer above, while Paths II and III offer the migrating sites without neighboring transition metals in both layers above and below (green and pink octahedrons, respectively). For three paths, either a negative or positive energy slope is possible, which is in contrast to the previous observations on the O2-type layered oxides with a superstructure that displayed an uphill reaction for further in-plane migration (**Figure R7a**).¹⁷ In O2-LLCO, if TM migrates along Path III, the results indicate that further in-plane TM migration is likely to occur, similar to typical O3- Li_2RuO_3 . The fussy return path would inevitably result in some migrated TMs remaining in the Li layer after the initial cycle (**Figure R7b**) compared with the pristine state (**Figure R7a**), thereby triggering the structural disordering and oxygen redox asymmetry even in O2-type layered oxides. In our O2-LLRO, due to the bigger size of Ru compared with Mn, it will receive a large energy obstacle when Ru ions pass through them. Instead, Ru ions are preferential to pass through OO path and stay in intermedia octahedron sites, the one-step return path makes the migrated Ru ions easily return to the initial sites, thereby increasing the reversibility of TM migration and oxygen redox symmetry.

As shown in previous reports,^{18, 19} the ratio of Li ions in TM layers will also affect the transition metal migration even in O2-type layered oxides with inherent structural robustness.

Figure R7. The calculation and experiment results of superstructure-free O₂-LLCO. ¹

4. How to understand the pattern in Figure 1f marked as “stacking faults”? Do the authors think the stacking faults are important to the electrochemical behavior?

Reply:

Stacking faults means that there are at least two types of interlacing arrangement within the layered structure. In our O₂-LLRO, stacking faults may be caused by the ion exchange process, during which the structure will transform from P2 to O₂ type.²⁰ Although the Li molten salts are excessive, we cannot exclude that there still exist a few P2-type Na-based residuals, which results in the coexistence of P2-type structure and O₂-type structure, that is “stacking faults”.

Zeng et al. revealed that stacking faults have a significant negative impact on Li ions diffusion in layered oxide cathode materials (**Figure R8**).²¹ They evidenced that the stacking fault in Li-rich layered oxides interrupts the straight out-of-plane migration and forces Li ions to take high-energy barrier diffusion pathways, which critically determines the sluggish diffusion kinetics of out-of-plane paths, resulting in poor rate performance. As a result, it is effective in increasing the electrochemical performance by reducing stacking faults.

Figure R8. The schematic diagram of the influence of stacking fault on the Li ions diffusion path. ²¹

5. The O2-type $\text{Li}_{0.6}\text{Li}_{0.2}\text{Ru}_{0.8}\text{O}_2$ and O3-type Li_2RuO_3 exhibit different discharge behavior, as shown in Figure 6d and Figure S16. How to understand the phenomenon?

Reply:

Upon discharge process, in general, the high voltage region corresponds to the reduction of oxygen anion, while the low voltage region corresponds to the reduction of Ru ions. According to the previous reports, the oxygen redox exhibits more sluggish kinetics than cations redox.^{12,14} Therefore, if the current density increases, there will be an obvious drop of the discharge capacity above 3.4 V for either O2-LLRO or O3- Li_2RuO_3 ; generally, the discharge capacity below 3.4 V would keep almost unchanged because the kinetics of cations redox is fast. However, for O3- Li_2RuO_3 (Figure S15), upon the current density increases, there is a significant decrease in both areas because the sluggish oxygen redox spreads into the low voltage region, meaning the asymmetry of anionic redox in O3- Li_2RuO_3 . While there is no obvious decrease for the discharge capacity below 3.4 V in O2-LLRO (Figure 6d), meaning more symmetry of anionic redox, which results from the reversible TM migration.

We hope that our responses meet your satisfaction and the manuscript is now suitable to be accepted by *Nature Communications*.

References:

1. Eum, D. et al. Effects of cation superstructure ordering on oxygen redox stability in O₂-type lithium-rich layered oxides. *Energy Environ. Sci.* **16**, 673-686 (2023).
2. Sathiya, M. et al. Reversible anionic redox chemistry in high-capacity layered-oxide electrodes. *Nat. Mater.* **12**, 827-835 (2013).
3. Sathiya, M., Leriche, J. B., Salager, E., Gourier, D., Tarascon, J. M., Vezin, H. Electron paramagnetic resonance imaging for real-time monitoring of Li-ion batteries. *Nat. Commun.* **6**, 6276 (2015).
4. Eric, M. et al. Visualization of O-O peroxy-like dimers in high-capacity layered oxides for Li-ion batteries. *Science* **350**, 1516-1521 (2015).
5. Qiao, Y. et al. Reversible anionic redox activity in Na₃RuO₄ cathodes: a prototype Na-rich layered oxide. *Energy Environ. Sci.* **11**, 299-305 (2018).
6. Li, X., Qiao, Y., Guo, S., Jiang, K., Ishida, M., Zhou, H. A New Type of Li-Rich Rock-Salt Oxide Li₂Ni_{1/3}Ru_{2/3}O₃ with Reversible Anionic Redox Chemistry. *Adv. Mater.* **31**, 1807825 (2019).
7. Li, X. et al. Direct Visualization of the Reversible O²⁻/O⁻ Redox Process in Li-Rich Cathode Materials. *Adv. Mater.* **30**, 1805197 (2018).
8. Ning, F. et al. Inhibition of oxygen dimerization by local symmetry tuning in Li-rich layered oxides for improved stability. *Nat. Commun.* **11**, 4973 (2020).
9. Satish, R. et al. Exploring the influence of iron substitution in lithium rich layered oxides Li₂Ru_{1-x}Fe_xO₃: triggering the anionic redox reaction. *J. Mater. Chem. A* **5**, 14387-14396 (2017).
10. Sim, R., Manthiram, A. Factors Influencing Gas Evolution from High-Nickel Layered Oxide Cathodes in Lithium-Based Batteries. *Adv. Energy Mater.* **14**, 2303985 (2024).
11. Liao, Y. et al. Electrolyte Degradation During Aging Process of Lithium-Ion Batteries: Mechanisms, Characterization, and Quantitative Analysis. *Adv. Energy Mater.*, DOI: 10.1002/aenm.202304295 (2024).
12. Assat, G., Foix, D., Delacourt, C., Iadecola, A., Dedryvère, R., Tarascon, J.-M. Fundamental interplay between anionic/cationic redox governing the kinetics and thermodynamics of lithium-rich cathodes. *Nat. Commun.* **8**, 2219 (2017).
13. Gent, W. E. et al. Coupling between oxygen redox and cation migration explains unusual electrochemistry in lithium-rich layered oxides. *Nat. Commun.* **8**, 2091 (2017).
14. Assat, G., Delacourt, C., Corte, D. A. D., Tarascon, J.-M. Editors' Choice—Practical Assessment of Anionic Redox in Li-Rich Layered Oxide Cathodes: A Mixed Blessing for High Energy Li-Ion Batteries. *J. Electrochem. Soc.* **163**, A2965-A2976 (2016).
15. Gregory Gachot et al. Gas Chromatography/Mass Spectrometry As a Suitable Tool for the Li-Ion Battery Electrolyte Degradation Mechanisms Study. *Anal. Chem.* **83**, 478-485 (2011).
16. Cao, X., Qiao, Y., Jia, M., He, P., Zhou, H. Ion-Exchange: A Promising Strategy to Design Li-Rich and Li-Excess Layered Cathode Materials for Li-Ion

- Batteries. *Adv. Energy Mater.* **12**, 2003972 (2021).
17. Eum, D. et al. Voltage decay and redox asymmetry mitigation by reversible cation migration in lithium-rich layered oxide electrodes. *Nat. Mater.* **19**, 419-427 (2020).
 18. Cui, C. et al. Structure and Interface Design Enable Stable Li-Rich Cathode. *J. Am. Chem. Soc.* **142**, 8918-8927 (2020).
 19. de Boisse, B. M., Jang, J., Okubo, M., Yamada, A. Cobalt-Free O2-Type Lithium-Rich Layered Oxides. *J. Electrochem. Soc.* **165**, A3630-A3633 (2018).
 20. Saïbi, V., Castro, L., Sugiyama, I., Belin, S., Delmas, C., Guignard, M. Stacking Faults in an O2-type Cobalt-Free Lithium-Rich Layered Oxide: Mechanisms of the Ion Exchange Reaction and Lithium Electrochemical (De)Intercalation. *Chem. Mater.* **35**, 8540-8550 (2023).
 21. Zeng, W. et al. Stacking Fault Slows Down Ionic Transport Kinetics in Lithium-Rich Layered Oxides. *ACS Energy Lett.* **9**, 346-354 (2024).

REVIEWER COMMENTS

Reviewer #1 (Remarks to the Author):

The reviewer thanks the authors for the responses to all comments and questions.

1. Adding the word “pristine” to the title looks awful to me. It feels worse and more confusing than the original title. As the authors also mentioned, the superstructure does not exist in the pristine state but also disappears in extended cycles, i.e., not only pristine. I did not disagree with the technical findings, but again, the importance on this point should be the different superstructure behavior between this material and others. Why don't the authors just add the main clarifications in their answer here to the manuscript? For example, as a brief summary of the superstructure discussion maybe.

2. The authors missed the point here. I'm familiar with the works the authors mentioned in the response, which is the whole point of this comment. The experimental results in this work (Raman) clearly show the oxidized oxygen is NOT O₂ gas molecule, this is inconsistent with the O₂ gas molecule (not O₂⁻) model that is heavily cited as Ref. 14 in this work. This is actually important to be pointed out clearly because it provides an important information for understanding the oxygen chemistry in such materials. But it seems nothing was changed to the manuscript. To me, the data here reveal an important disagreement with the central conclusion of Ref. 14 and other works, which naturally enhances the novelty of this work.

Reviewer #2 (Remarks to the Author):

In the revised manuscript, the authors managed to strengthen their claims. The thorough revision of the manuscript has addressed all of my concerns and has significantly improved its value for the community. I recommend publication as is.

Reviewer #3 (Remarks to the Author):

The authors have well addressed the reviewer's comments and the manuscript can be accepted now.

NCOMMS-23-61143A
RESPONSE LETTER TO REVIEWERS

Reviewer #1:

1. Adding the word “pristine” to the title looks awful to me. It feels worse and more confusing than the original title. As the authors also mentioned, the superstructure does not exist in the pristine state but also disappears in extended cycles, i.e., not only pristine. I did not disagree with the technical findings, but again, the importance on this point should be the different superstructure behavior between this material and others. Why don't the authors just add the main clarifications in their answer here to the manuscript? For example, as a brief summary of the superstructure discussion maybe.

Reply: We sincerely appreciate your advice to improve the quality of our manuscript.

According to your advice, we have now deleted the “pristine” in the title and added the relative description in the revised manuscript: Although the structure of O2-LLRO during the initial charge process evolves into the superstructure, which may be significantly associated with the out-of-plane migration of Li_{TM} ions. It should be emphasized that the superstructure does not exist in the important pristine state, the initial discharged state and thereafter the extended cycles. For the reasons, the O2-LLRO is regarded as a superstructure-free material.

2. The authors missed the point here. I'm familiar with the works the authors mentioned in the response, which is the whole point of this comment. The experimental results in this work (Raman) clearly show the oxidized oxygen is NOT O₂ gas molecule, this is inconsistent with the O₂ gas molecule (not O²⁻) model that is heavily cited as Ref. 14 in this work. This is actually important to be pointed out clearly because it provides an important information for understanding the oxygen chemistry in such materials. But it seems nothing was changed to the manuscript. To me, the data here reveal an important disagreement with the central conclusion of Ref. 14 and other works, which naturally enhances the novelty of this work.

Reply: We thank for your advice to improve the novelty of the manuscript.

According to your advice, we have now added the relative description in the revised manuscript: The oxygen evolution process in O2-LLRO is quite different from that of Na_{0.6}Li_{0.2}Mn_{0.8}O₂ with ribbon-ordered superstructure and Na_{0.75}Li_{0.25}Mn_{0.75}O₂ with honeycomb-ordered superstructure reported in previous work,¹ in which the oxygen evolution species in two samples can only be detected as trapped molecular O₂. The discrepancy highlights the importance of the pristine structure, which may directly affect the oxygen evolution path.

We hope that our responses meet your satisfaction and the manuscript is now suitable to be published by *Nature Communications*.

Reference:

1. House, R. A. et al. Superstructure control of first-cycle voltage hysteresis in oxygen-redox cathodes. *Nature* **577**, 502-508 (2020).

REVIEWERS' COMMENTS

Reviewer #1 (Remarks to the Author):

Thanks for adopting the suggestions. Congratulations!